# Risk Assessment Based on Nitrogen and Phosphorus Forms in Watershed Sediments: A Case Study of the Upper Reaches of the Minjiang Watershed

Hongmeng Ye [1,2,3], Hao Yang [1], Nian Han [3], Changchun Huang [1], Tao Huang [1], Guoping Li [2], Xuyin Yuan [2,3,*] and Hong Wang [1,*]

[1]  School of Geography Science, Nanjing Normal University, Nanjing 210023, China; hongmengye@sina.com (H.Y.); yanghao@njnu.edu.cn (H.Y.); huangchangchun_aaa@163.com (C.H.); huangtao198698@126.com (T.H.)

[2]  Fujian Provincial Key Laboratory of Eco-Industrial Green Technology, College of Ecology and Resource Engineering of Wuyi University, Wuyishan 354300, China; ptlgp@126.com

[3]  College of Environmental, Hohai University, Nanjing 210098, China; hannian_hhu@163.com

*  Correspondence: yxy_hjy@hhu.edu.cn (X.Y.); hongwang@njnu.edu.cn (H.W.)

**Abstract:** In order to achieve effective eutrophication control and ecosystem restoration, it is of great significance to investigate the distribution characteristics of nutrient elements in sediments, and to perform ecological risk assessments. In the current grading criteria for nutrient elements in sediments, only the overall or organic components of carbon, nitrogen and phosphorus are considered, while the specific species distributions and bioavailability characteristics are rarely taken into account. Hence, using the current grading criteria, the differences in the release, migration and biological activity of nutrient elements in sediments cannot be accurately reflected. Taking the upper reaches of the Minjiang River watershed as an example, we analyzed the overall distributions and the ratio of nutrient elements in sediments, the spatial changes of nitrogen and phosphorus forms, the bioavailability, and the environmental significance. The ecological risk of nitrogen and phosphorus in sediments was assessed using an evaluation method based upon the biological effective parameter. The results were compared with the results of the evaluation methods based on the single pollution index, and then these evaluation methods were confirmed accordingly. From the results, the following conclusions can be obtained: (1) The spatial distributions of nutrient elements in sediments in the upper reaches of the Minjiang River Watershed (including the Jianxi Basin, Futunxi Basin, and Shaxi Basin) were significantly affected by the local ecology and the urban sewage discharge system. (2) The maximum average contents of total organic carbon (TOC), total nitrogen (TN), and total phosphorus (TP) in sediments were observed in the Jianxi Basin, the Futunxi Basin and the Shaxi Basin, respectively. (3) According to the contents of nitrogen and phosphorus in sediments, the bioavailable nitrogen (TTN) accounted for 35.49% of the total contents of TN. The components of TTN can be sorted from high to low as follows: Nitrogen in organic sulfide form (SOEF-N) > nitrogen in iron-manganese oxide form (SAEF-N) > nitrogen in ion exchange form (IEF-N) > nitrogen in weak acid leaching form (WAEF-N). Inorganic phosphorus (IP) was the main component of TP. The components of IP can be sorted from high to low as follows: Metal oxide bound phosphorus (NaOH-P) > calcium bound phosphorus (HCl-P) > reduced phosphorus (BD-P) > weakly adsorbed phosphorus ($NH_4Cl$-P). Meanwhile, bioavailable phosphorus (BAP, BAP = $NH_4Cl$-P + BD-P + NaOH-P) accounted for 36.94% of TP. According to the results of the single pollution index method, the risk level of TOC pollution in the sediments was relatively low in the whole area, while the risk level of TN pollution was low or moderate in most zones, and severe in certain ones. The risk level of TP pollution was low to moderate. (4) From the results of the bioavailability index evaluation method, based on the total amounts and forms of N and P, the risk level of N pollution was moderate, while the risk of P pollution was negligible. In addition, the results of the bioavailability index evaluation method were

more consistent with the actual situation and reflected the overall environmental effects of nitrogen and phosphorus.

**Keywords:** sediment; nutrient element; nitrogen forms; phosphorus forms; risk assessment

---

## 1. Introduction

In recent years, eutrophication has become one of the most serious problems in the water environment and water sediments, and it has attracted increasing attention globally [1,2]. Sediment can store the persistent pollutants, as well as absorb or release water pollutants. Upon the settlement of particles to the bottom, the environmental information is preserved in the sedimentary environment. The source elements, i.e., carbon, nitrogen, phosphorus, etc., in the sediment, can reflect the changes of the nutrient process, primary productivity, and the pollution sources of rivers [3–5]. However, the distribution of nutrient elements in sediments is affected by various factors. The biological effects of nutrient elements in sediments are mostly related to the biological effects of the polluted areas. In most cases, sediment pollution has no significant effect on biological communities and aquatic ecosystems. The concentrations of nutrient elements in sediments are not necessarily proportional to their ecological toxicity levels. Therefore, compared with the environmental quality assessment in water, the environmental quality assessment in sediment is more complicated and uncertain.

Grading criteria, including sediment quality assessment guidelines (by the Department of Environment and Energy, Ontario, Canada), the single pollution index, overall pollution index, organic index and organic nitrogen index have been widely used for the assessment of sediment nutrient elements [5–11]. However, using the above indices, only the total or organic components of carbon, nitrogen and phosphorus are considered, while the chemical species and bio-availability are rarely taken into account. Thus, these indices cannot accurately reflect the nutrient release, migration and biological activity in sediments. Since bio-availability and environmental geochemical behaviors are related to the forms of nitrogen and phosphorus in sediments [12,13], there is an urgent need for a comprehensive assessment method based on the biological effective index, considering both the total amount of nitrogen and phosphorus and their ecologically bioavailable fractions. Accurate and effective evaluation of the comprehensive eco-environmental effects of nutrient elements in sediments is of great significance. Specifically, scientific evaluation of the ecological risk of nutrient elements in sediments is particularly important for forestry mountainous basins, due to their heterogeneity of land use and the complexity of their topography.

The Minjiang River is the largest river in Fujian Province, whose upper reaches account for nearly 70% of its total Basin area. Hence, the accumulation of nutrient elements in sediments in this area is a key factor for the risk assessment, eutrophication control and water quality control of the Minjiang River [14]. It has been reported that the water body in the upper reaches of the Minjiang River is in a state of being oligotrophic to mesotrophic, in which nitrogen and phosphorus are the main pollutants [15]. However, only a few studies on the distributions and risk assessments of nutrient elements in the sediment of this mountainous watershed have been reported.

In this paper, taking the upper reaches of the Minjiang River as an example, the total distributions and ratios of nutrient elements, the spatial changes of nitrogen and phosphorus forms, and the bioavailability in sediments, were analyzed. In addition, based upon our analyses, an index was proposed to evaluate the ecologically bioavailable fractions of nitrogen and phosphorus, and to assess the ecological risks of these sediments. The evaluations results of using both the previous single pollution index [6,7] and the proposed index were compared, and the corresponding risk index evaluation method was determined accordingly. This study proposed a method to effectively assess the ecological risk of nutrient elements in watershed sediments, determine the control sources and

influencing factors of nutrient elements in sediments, and prevent eutrophication and ecological restoration in the watershed.

## 2. Materials and Methods

### 2.1. Overview of the Minjiang River

The Minjiang River is the "mother river" to the Fujian Province in China. It has a total length of 562 km (349 miles) and a watershed area of 60,992 km$^2$ (23,549 sq. mi.). Indeed, the watershed area accounts for 62.5% of the total area of Fujian. The basin above Nanping City is regarded as the upper reaches of the Minjiang River. This area accounts for nearly 70% of the entire Basin area. The annual runoff in this area is 1056.7 mm (41.6 "), and the water volume in this area accounts for 75% of the total water volume of the Minjiang River [16]. Therefore, the upper reaches of the Minjiang River have a great impact upon the ecological environment of the Minjiang River Basin, even the whole area of Fujian Province. Additionally, the upstream river is a typical mountain river with three main tributaries, i.e.: The Jianxi River, Futunxi River and Shaxi River. Among the tributaries, the Jianxi River is the north source of Minjiang River, which consists of the Chongyangxi River, Nanpuxi River, and Songxi River, and flows through Wuyishan, Pucheng, Songxi, Zhenghe, Jianyang and Jian'ou. It is called the Jianxi River after the confluence at Jian'ou, and finally flows to Nanping. The total length of the main stream of Jianxi River is 295 km (183 miles), and the average slope is 0.8‰. The Futunxi River is another source of the Minjiang River, and consists of Yanping, Guangze, Shaowu, Shunchang, Jianning, Taining and Jiangle. This river is divided into two branches above Shunchang, i.e., the north branch and the west branch. The north branch is the Futunxi River (length = 228 km, or 141.6 miles) and the west branch is Jianxi River (length = 253 km, or about 157 miles). Futunxi River has an average slope of 1.2‰. The Shaxi River (length = 328 km, or 203.8 miles) is the mainstream of the Minjiang River, and the average slope is 0.8‰. The Shaxi Basin involves Ninghua, Qingliu, Liancheng, Yong'an, Mingxi, Sanming, and Shaxian, as shown in Figure 1a.

The upper reaches of the Minjiang River are a subtropical monsoon climate zone with typical southern hilly landforms. The main lithology includes metamorphic sandstone, siliceous rock, siltstone and phyllite. The main soil types are red soil and yellow soil formed by granite and sandy rock. In recent years, due to economic development and urbanization, the industry and agriculture in the Minjiang River Basin have been developed rapidly, forming significantly expanded animal husbandry and aquaculture. The farmland runoff, domestic sewage, livestock and aquaculture wastewater have been discharged to the river along the coast, resulting in an increasing environmental load to the basin. The ecological function of the water body has been obviously reduced, which directly affects the ecological security and the sustainable development of the basin. From the reports, the upper reaches of the Minjiang River have gradually changed from a low nutrition state to a medium nutrition state, which is a critical index for water environment control. Excessive concentrations of nitrogen and phosphorus are the main causes of water pollution in the upper reaches of the Minjiang River [15]. According to the land use and coverage classification system of the Chinese Academy of Sciences, the spatial distribution of land use in the upper reaches of Minjiang River was extracted in 2015, and the distribution of land use types in the upper reaches of this Minjiang River was obtained (Figure 1b). The land use data were from the Resource and Environment Science Data Center of the Chinese Academy of Sciences (http://www.resdc.cn). According to the spatial distribution of land use, the percentages of woodland (natural and planted forests with canopy density of more than 30%), sparse woodland (forest with canopy density of less than 30%) and grassland were 47.28%, 15.78% and 14.55%, respectively. The percentages of paddy, dry land, and orchard were 11.79%, 5.11% and 3.70%, respectively. The percentage of other land use types was less than 1%.

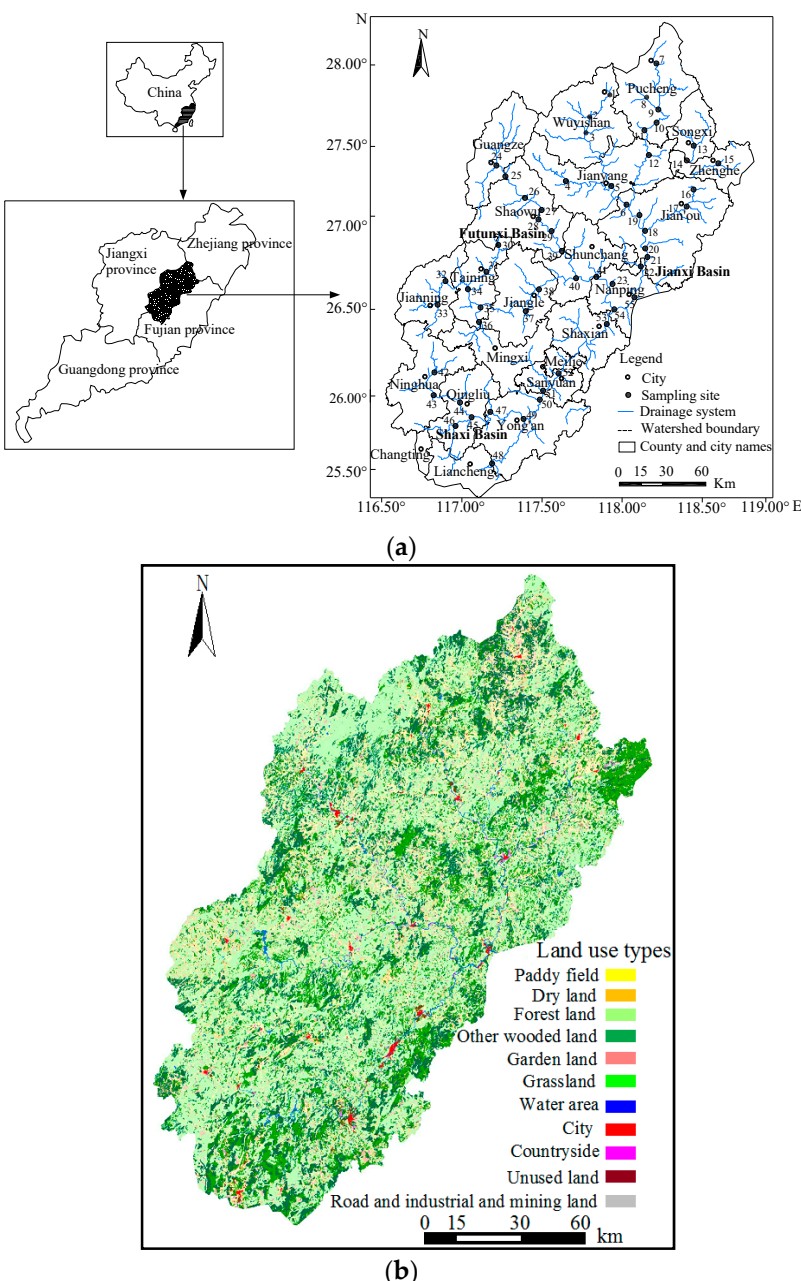

**Figure 1.** Administrative division and distribution of land use type in the upstream Watershed of the Minjiang River. (**a**) Administrative division and sediment sampling points in the upper reaches of the Minjiang River, (**b**) Distribution of land use types.

## 2.2. Sample Collection and Characterization

### 2.2.1. Sample Collection

In October and November 2016, 55 sediment samples were collected using a Beeker (Holand) sampling device with a cylinder (polyvinyl chloride (PVC) material) tube with a length of 100 cm and a diameter of 5 cm. The surface sediment samples with depths from 0 to 20 cm were mixed for sample storage and testing. Among them, the sampling points numbered 1 to 23 were located in Jianxi, the sampling points numbered 24 to 41 were located in Futunxi, and the sampling points numbered 42 to 55 were located in Shaxi, respectively, as shown in Figure 1a. Based on both GPS data and local record, the longitudes and latitudes of the sampling points, the administrative scope, the land use,

and human activities in local areas were recorded. These samples were freeze-dried, homogenized, ground, sieved to 0.25 mm and stored at 4 °C in plastic bags.

### 2.2.2. Sample Characterization

The particle size, pH value, total organic carbon (TOC), total nitrogen (TN), total phosphorus (TP) and other major elements (e.g., Fe, Mn, Al, Si, Ca) of the sediment samples were characterized. The pH value was determined by the electrode method. Organic carbon and nitrogen were oxidized by hydrated hot potassium dichromate and potassium sulfate, respectively. TP was measured by the alkali melting molybdenum antimony resistance spectrophotometry [17]. The particle size distributions of the sediments including clay (<4 μm), silt (4~64 μm), and sand fractions (>64 μm) were determined by the Mastersizer 2000 (Malvern, UK) [13,15]. The major elements (Ca, Al, Fe, Mn and Si) were determined using X-ray fluorescence spectrometer (PW2440, Philips, Almelo, The Netherlands).

The nitrogen forms in sediments were tested by the continuous extraction and fractionation of nitrogen [18]. Nitrogen can be divided as transformable nitrogen (TTN) and non-transformable nitrogen (NTN). TTN can be recycled in suitable environments, and thus is considered bioavailable. On the other hand, NTN exhibits relatively stable morphology during early diagenesis and does not participate in a short period of time [12,18]. Therefore, the TTN extracted by this method is the extractable nitrogen, which is called bioavailable nitrogen, and plays a key role in the geochemical behavior of nitrogen [12]. TTN can be further divided into nitrogen in ion exchange form (IEF-N), nitrogen in weak acid leaching form (WAEF-N), nitrogen in iron manganese oxide form (SAEF-N) and nitrogen in organic sulfide form (SOEF-N). The extraction processes for the fractions of TTN are shown in Figure 2a. The contents of $NH_4^+$-N and $NO_3^-$-N in the supernatant fluid were determined using the previously reported methods (Wang et al., 2009) [12]. In order to ensure the accuracy, each measurement was repeated three times to obtain the average.

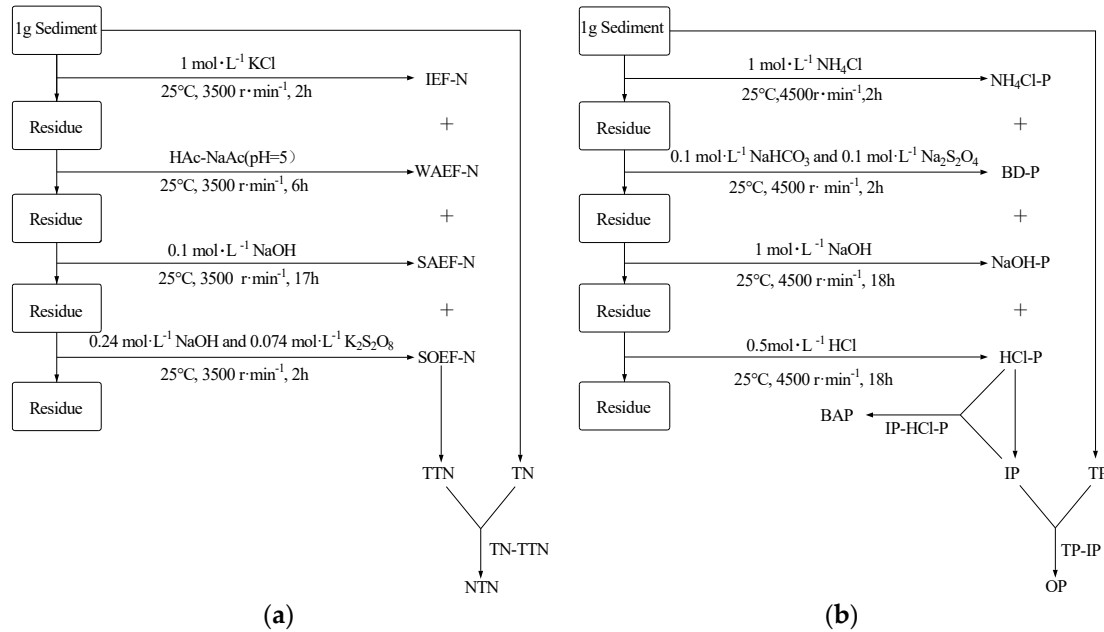

(**a**)　　　　　　　　　　　　　　　　　　　(**b**)

**Figure 2.** The extraction processes of nitrogen and phosphorus in sediments. (**a**) The extraction processes of nitrogen, (**b**) The extraction processes of phosphorus.

TP in sediments can be divided into inorganic phosphorus (IP) and organic phosphorus (OP). IP can be divided into weakly adsorbed phosphorus ($NH_4Cl$-P), reduced phosphorus (BD-P), metal oxide bound phosphorus (NaOH-P) and calcium bound phosphorus (HCl-P) [13,19]. Among them, $NH_4Cl$-P, BD-P and NaOH-P are the bioavailable forms of phosphorus (where BAP = $NH_4Cl$-P + BD-P + NaOH-P = IP-HCl), while HCl-P primarily comes from clastic rocks and autogenous sources [13,15].

Indeed, HCl-P is the most stable form and can be regarded as a permanent sink of phosphorus. In other words, IP is difficult to be utilized by organisms [13,19,20]. The extraction processes of phosphorus are shown in Figure 2b. Each phosphorus fraction was quantitatively assessed by the molybdenum blue/ascorbic acid method [13,15]. The experiments were repeated three times and the relative deviations of the results were controlled below 5%.

*2.3. Research Methods*

All reagents used in this study were of analytical grade. All solutions were prepared with deionized water (>18 MΩ). Meanwhile, all of the glassware and plastic ware were soaked in 0.3% HCl solution overnight, cleaned, and then rinsed three times with deionized water. A quality control procedure was applied for the sample preparation and analysis steps. In order to ensure the accuracy, each measurement was repeated three times to obtain the average. More detailed information was provided in the literature [12,13]. In addition, all the samples in these analyses were measured three times, and the error was controlled below 10%. The data was analyzed with SPSS 19.0 (IBM, New York, NY, USA), while charts were plotted using Origin 9.0 (Origin Lab, Northampton, MA, USA). In the statistical analysis, if $p < 0.05$, the differences in the total concentration, nitrogen forms, and phosphorus forms among the sediments from different sample sites were considered significant.

2.3.1. Risk Assessment Method

The ecological risks of nutrient elements in sediments collected from the target watershed were assessed by the traditional evaluation method based on a single pollution index and the evaluation method based on biological effective indices of nitrogen and phosphorus. The latter method was developed based on the former one, as discussed in the following section.

(1) Evaluation method based on a single pollution index

Currently, there are no universal standards and methods for the ecological risk assessment of nutrient elements such as nitrogen and phosphorus in river sediments. The widely used assessment method based on the single pollution index was developed based on the guidelines for environmental quality assessment formulated by the Department of Environment and Energy of Ontario, Canada (1992). This concise and simple method has been widely used in China [7].

The relationship is expressed as:

$$Pi = Ci/Cs \tag{1}$$

where *Pi* is the single evaluation index or standard index, *Ci* is the measured concentration of the evaluation factor *i* (*i* is TOC, TN or TP), and *Cs* is the standard concentration of this evaluation factor *i*. Based on the regulations of the safe concentration limits for the nutrient elements in aquatic organisms in the Sediment Quality Guidelines (SQGs) [6], the standard concentrations of TOC, TN and TP should be 1%, 550 mg·kg$^{-1}$ and 600 mg·kg$^{-1}$, respectively. According to the value of *Pi* index, the risks can be divided into four levels, as shown in Table 1.

**Table 1.** Evaluation criterion of nutrient elements in sediments.

| Risk Level | TOC | TN | TP | Pollution Assessment |
|---|---|---|---|---|
| I | $P_{TOC} < 1$ | $P_{TN} < 1$ | $P_{TP} < 0.5$ | Clean |
| II | $1 \leq P_{TOC} < 5$ | $1 \leq P_{TN} < 2$ | $0.5 \leq P_{TP} < 1$ | Slightly polluted |
| III | $5 \leq P_{TOC} < 10$ | $2 \leq P_{TN} < 3$ | $1 \leq P_{TP} < 1.5$ | Moderately polluted |
| IV | $10 \leq P_{TOC}$ | $3 \leq P_{TN}$ | $1.5 \leq P_{TP}$ | Seriously polluted |

(2) The evaluation method based on biological effective indices of nitrogen and phosphorus

The bioavailability coefficients of nitrogen (N) and phosphorus (P) in sediments were proposed on the basis of (i) the evaluation method based on the single pollution index and (ii) the distribution

ratio of secondary phase to primary phases. In the proposed method, the effects of total and individual contents of N and P in all forms were considered. Specifically, the RSP (Ratio of Secondary Phase to Primary Phase) evaluation method was used to evaluate the contents ratio between the secondary phase and primary phase. This method was generally used to distinguish the sources of heavy metals (natural or artificial sources) and reflect the chemical activity and bioavailability of heavy metals [21].

In terms of the extraction sequence and bioavailability of N (or P), the evaluation method based on a single pollution index was modified. The ratio of transformable N (or bioavailable P) to non-transformable N (or non-bioavailable P) was defined as the bioavailable coefficient of N (or P). Therefore, this method is also called the bioavailable index evaluation method of N and P. The equations are described as follows:

$$PKi = Ki \cdot Pi = Ki \cdot Ci/Cs \tag{2}$$

$$K_N = C_{TTN}/(C_{TN} - C_{TTN}) \tag{3}$$

$$K_P = C_{BAP}/(C_{TP} - C_{BAP}) \tag{4}$$

where *PKi* is the biological effective index of N (or P), *Ki* is the biological efficiency coefficient of factor *i* (*i* refers to *N* or *P* here). $K_N$ is the biological efficiency coefficient of N, and $K_P$ is the biological efficiency coefficient of P. *Pi, Ci* and *Cs* are defined the same as above. $C_{TN}$, $C_{TTN}$, $C_{TP}$, $C_{BAP}$ are the standard concentrations of TN, TTN, TP and BAP, respectively. The risk levels of biological effective index *PKi* are defined the same as that (*Pi*) in the evaluation method based on single pollution index.

### 2.3.2. Analysis Method of Factors Affecting Distributions of Nitrogen and Phosphorus Forms

In this study, the effects of the basic physicochemical parameters on N and P forms were investigated by the correlation coefficients between the concentration of basic physicochemical indices (e.g., pH, particle size, TOC, TN, TP, Fe, Mn, Al, Si and Ca) and the concentrations of N and P forms in sediments. Using the Principle component analysis (PCA) method, the effects of natural and human factors on N and P forms in sediments were identified and studied through dimension reduction of different physicochemical indicators. The correlation coefficients and PCA analysis were performed using the SPSS 19.0 [13,19].

## 3. Results and Discussion

### *3.1. Spatial Distributions of Overall Nutrient and Sediment Morphology*

#### 3.1.1. Distributions of Total Nutrient Elements and Ratio of Nutrient Elements in Sediments

The overall distribution of nutrient elements in sediments of the upper reaches of Minjiang River is shown in Table 2. The contents of TOC, TN and TP in sediment of the upper reaches were 1.11–3.05%, 643.28–2427.82 mg·kg$^{-1}$ and 190.09–722.60 mg·kg$^{-1}$, respectively. From the distribution results, the mean value of TOC in the sediments was in the order of high to low in the following sub-watersheds: Jianxi Basin > Shaxi Basin > Futunxi Basin. Similarly, the orders of sub-watersheds for the mean value of TN and TP contents were Futunxi Basin > Shaxi Basin > Jianxi Basin and Shaxi Basin > Futunxi Basin > Jianxi Basin, respectively. The difference in the contents of the same element in the sediments from the three basins indicated the variation of element accumulation origin among the basins. Meanwhile, the difference in the content among different elements can reflect the distinction of the main element sources in the basin. Indeed, the distributions of nutrient elements are related to the geological background, geographical environment, land use, human activities and water conditions [22,23].

**Table 2.** Distributions of total nutrient elements in sediments.

| Basin | Items | Max. | Mean | Min. | SD. | CV (%) |
|---|---|---|---|---|---|---|
| Jianxi | TOC (%) | 3.05 | 1.85 [a] | 1.21 | 0.53 | 28.74 |
| | TN (mg·kg$^{-1}$) | 1440.59 | 1083.37 [b] | 754.88 | 218.86 | 20.20 |
| | TP (mg·kg$^{-1}$) | 673.59 | 431.65 [b] | 190.09 | 122.32 | 28.34 |
| Futunxi | TOC (%) | 2.82 | 1.61 [a] | 1.15 | 0.48 | 29.50 |
| | TN (mg·kg$^{-1}$) | 2427.82 | 1439.48 [a] | 643.28 | 457.15 | 31.76 |
| | TP (mg·kg$^{-1}$) | 689.42 | 513.35 [a,b] | 266.62 | 125.24 | 24.40 |
| Shaxi | TOC (%) | 2.37 | 1.81 [a] | 1.11 | 0.33 | 17.93 |
| | TN (mg·kg$^{-1}$) | 2038.29 | 1333.36 [a] | 812.51 | 348.43 | 26.13 |
| | TP (mg·kg$^{-1}$) | 722.60 | 540.44 [a] | 249.87 | 152.12 | 28.15 |

Note: Different letters ([a] to [b]) implicate the significant difference between locations.

Previous studies have shown that the ratios of nutrient elements in sediments can reflect the geochemical behaviors of elements, the environment and the internal and external pollution sources [9]. Specifically, the TOC/TN ratio is widely used to identify the potential sources of organic matters and the variations between species. The TOC/TP ratio reflects the decomposition rate of organic carbon and phosphorus compounds in sediments to a certain extent [7]. The TN/TP ratio reflects the dynamic processes of the accumulation, deposition and release of nitrogen and phosphorus in water [5,7]. If TOC/TN > 10, the organic matter is primarily from land sources; if TOC/TN = 10, the internal and external organic matters are basically in equilibrium; if TOC/TN < 10, the organic matter is primarily from water bodies [5,7].

From Table 3, in the upper reaches of the Minjiang River, the TOC/TN ratios in sediments ranged from 6.39 to 32.30. In addition, the average TOC/TN ratios of sediments in Jianxi, Futunxi and Shaxi Basins were 17.78, 12.23 and 14.27, respectively. The TOC/TN ratios of sediments in the upper reaches of the Minjiang River were higher than the TOC/TN ratio in the wetland soils (ranged 8.0–15.5 with an average of 11.7), the TOC/TN ratio in the sediments in the lower reaches of Minjiang River (ranged 10.3–13.2 with the average of 11.9) [7], and the average TOC/TN ratio in China (11.9) [24]. Due to the high forest coverage (more than 75%), large rainfall and terrain gradient in the upper reaches of the Minjiang River, plant litter and debris in this area are readily flushed into the river during soil erosion. Therefore, the TOC accumulation and TOC/TN ratio in sediments in the upper reaches of the Minjiang River are positively related to the organic components from terrestrial sources [9].

The TOC/TP ratios in sediments in the upper reaches of the Minjiang River ranged from 16.58 to 103.39. The average values of the TOC/TP ratios in the Jianxi, Futunxi and Shaxi Basins were 45.97, 33.22 and 36.57, respectively, which were significantly higher than that in the wetland soils in the Minjiang Estuary (which ranged from 16.0 to 56.1, with an average of 27.4) [7]. On the one hand, the spatial variations of nutrient element ratios in sediments in the upstream basin are relatively large. On the other hand, land use in the upstream basin has a significant impact on the spatial variations of nutrient element ratios, because compared with the downstream basin, the upstream basin has higher forest land coverage and lower urbanization. Therefore, the accumulation of TOC in the upstream watershed dominates the TOC/TP ratio.

The TN/TP ratios in sediments in the upper reaches of Minjiang River ranged from 1.22 to 5.12, and the average values in the Jianxi, Futunxi and Shaxi Basins were 2.67, 2.93 and 2.55, respectively. In addition, the TN/TP ratio in sediments in the upper reaches was higher than that of wetland soils in the Minjiang Estuary (this ranged from 1.5 to 4.2 with an average of 2.3) [7]. This phenomenon was related to the severe farmland pollution in the upstream basin and the severe municipal pollution in the downstream basin [15].

**Table 3.** Total organic carbon/total nitrogen (TOC/TN), total organic carbon/total phosphorus (TOC/TP) and total nitrogen/total phosphorus (TN/TP) ratios in sediments.

| Basin | Items | Max. | Mean | Min. | SD. | CV (%) |
|---|---|---|---|---|---|---|
| | TOC/TN | 32.30 | 17.78 | 10.00 | 6.50 | 36.57 |
| Jianxi | TOC/TP | 103.39 | 45.97 | 24.53 | 18.47 | 40.19 |
| | TN/TP | 4.09 | 2.67 | 1.54 | 0.72 | 26.94 |
| | TOC/TN | 24.73 | 12.23 | 6.39 | 4.85 | 39.68 |
| Futunxi | TOC/TP | 57.56 | 33.22 | 18.53 | 11.48 | 34.54 |
| | TN/TP | 5.12 | 2.93 | 1.22 | 0.99 | 33.63 |
| | TOC/TN | 19.28 | 14.27 | 8.45 | 3.40 | 23.85 |
| Shaxi | TOC/TP | 66.69 | 36.57 | 16.58 | 12.29 | 33.61 |
| | TN/TP | 3.46 | 2.55 | 1.93 | 0.50 | 19.57 |

### 3.1.2. Spatial Distribution of Nitrogen in Sediments

The distribution of bioavailable nitrogen in sediments in the upper reaches of the Minjiang River is shown in Figure 3. As observed, the spatial distribution and concentration of nitrogen varied significantly among different sediments. The TTN concentration ranged from 251.18 to 976.59 mg·kg$^{-1}$, accounting for 22.18% to 55.49% of TN content.

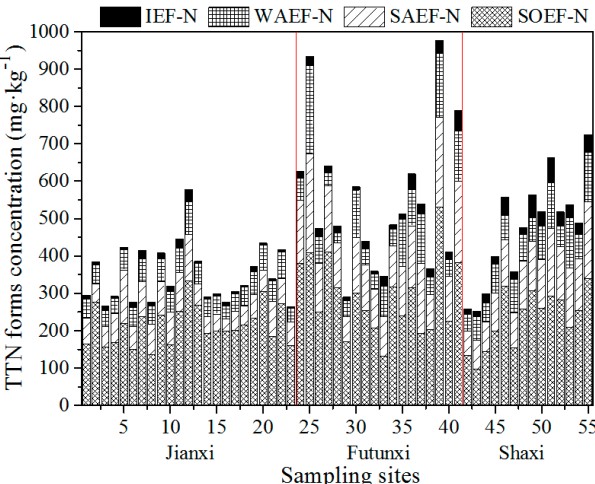

**Figure 3.** Concentration and spatial distributions of transformable nitrogen (TTN) in sediments. Note: The red line is the limit of sampling points in different watershed.

The maximum TTN concentration was observed in Samples No. 25 and No. 39. Sample No. 25 was located in the suburb of Guangze County in the upper reaches of the Futunxi River. The sampling site was close to residential areas, thus untreated discharges of livestock and poultry manure, domestic refuse and domestic sewage were found in the sample. Sample No. 39 was located in the suburb of Shunchang County in the lower reaches of the Futunxi River. The sampling site was close to farmland, houses, sand quarries and wastewater. The minimum TTN concentration was observed in Samples No. 42 and No. 43, both of which were collected in the suburb of Ninghua County in the upper reaches of the Shaxi River Basin. Both sampling sites were close to farmland and woodland. From the results, the intensity of land use and human activities had a significant effect on the the accumulation of TTN in sediments [12,18].

In order to further investigate the distribution of nitrogen forms, the distributions of TTN in sediments in the three sub-basins are summarized in Figures 4 and 5. The average TTN/TN ratio in Futunxi (39.15%) was slightly higher than that in the Jianxi (32.76%) and the Shaxi (35.26%). Meanwhile, the TTN concentration in sediments in the Futunxi Basin (548.18 mg·kg$^{-1}$) was higher than that of

the Jianxi (351.19 mg·kg$^{-1}$) and Shaxi (472.19 mg·kg$^{-1}$) Basins. Therefore, the biological availability of nitrogen was maximum in the sediments in the Futunxi Basin.

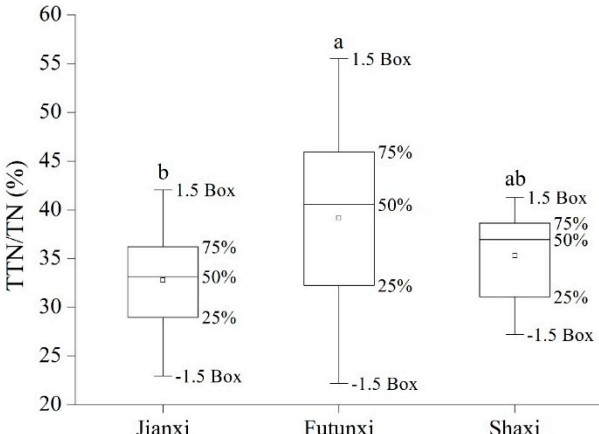

**Figure 4.** Percentage of bioavailable, transformable nitrogen (TTN) to total nitrogen (TN) in sediments. Different letters (a to b) implicate the significant difference between locations.

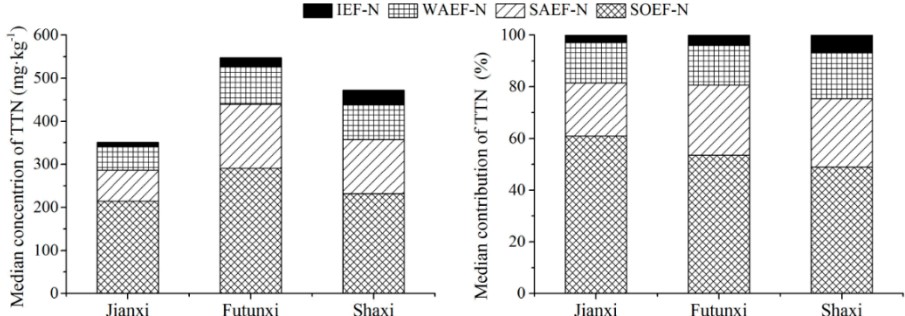

**Figure 5.** Contents and percentages of TTN in sediments.

The average concentrations of IEF-N in sediments in the Jianxi, Futunxi and Shaxi Basins were 10.31 mg·kg$^{-1}$, 21.76 mg·kg$^{-1}$ and 33.94 mg·kg$^{-1}$, respectively. Meanwhile, the IEF-N/TTN ratios in the Jianxi, Futunxi and Shaxi Basins were 2.85%, 4.01% and 6.93%, respectively. Both the IEF-N concentrations and IEF-N/TTN ratios were sorted from high to low in the following areas: Shaxi > Futunxi > Jianxi. Previous studies have shown that IEF-N in sediments may be an indicator of nitrogen mobilities and their availability for plant uptake [12,18]. Indeed, IEF-N accounted for over 50% of nitrogen released from the sediment to water [25]. Therefore, nitrogen in sediments was readily released in the Shaxi River Basin.

WAEF is carbonate-bound nitrogen, and its release capacity is only inferior to that of IEF-N. The production and distribution of WAEF are mainly affected by the carbonate content and pH of soil [18]. In an acidic environment (low pH), WAEF is readily transferred from sediment to interstitial water, resulting in a decreased content in the sediment. The WAEF-N/TTN ratios in these Jianxi, Futunxi and Shaxi sediments were 15.59%, 15.35% and 17.58%, respectively. Therefore, the distribution of the WAEF-N/TTN ratio was relatively stable.

SAEF-N, also known as ferromanganese oxidized nitrogen, is primarily Fe/Mn oxides-bound nitrogen. The content and distribution of SAEF-N are mainly related to the redox potential and the content of metal minerals in the deposition environment. Under reduction conditions, SAEF-N is readily released from the sediments in the interstitial water [12]. In addition, the SAEF-N/TTN ratios in sediments (average = 20.60–27.15% in each basin) were significantly larger than that in the first two components. The contents of SAEF-N in sub-basins ranged from 72.33 to 148.64 mg·kg$^{-1}$.

SOEF-N is organic sulfide-bound nitrogen, in which nitrogen is primarily bound to non-degradable macromolecule organic matter or sulfide. SOEF-N exhibits stable properties [25]. Previous studies have shown that SOEF-N has the minimum TTN strength but high concentrations. The contribution of SOEF-N to the release of nitrogen from the sediment to the water body is only inferior to that of IEF-N [12,18,25]. The concentrations of SOEF-N in sediments were ordered from high to low as follows: Futunxi (290.70 mg·kg$^{-1}$) > Shaxi (232.20 mg·kg$^{-1}$) > Jianxi (214.17 mg·kg$^{-1}$). In addition, the SOEF-N/TTN ratio was ordered from high to low as follows: Jianxi (60.96%) > Futunxi (53.49%) > Shaxi (48.91%).

In summary, the concentrations of the bioavailable nitrogen contents in sediments in the upper reaches of the Minjiang River were in the order of SOEF-N > SAEF-N > WAEF-N > IEF-N. Among them, the content of SOEF-N was above 50% of the total bioavailable nitrogen, and considered as the dominant component of bioavailable nitrogen.

The composition of inorganic phosphorus (IP) in the sediments of the upper reaches of the Minjiang River is shown in Figure 6. As observed, the composition and concentration distributions of phosphorus were highly heterogeneous among different sediments. The overall IP concentration ranged from 125.81 mg·kg$^{-1}$ to 615.40 mg·kg$^{-1}$, accounting for 42.80% to 89.26% of the overall TP content.

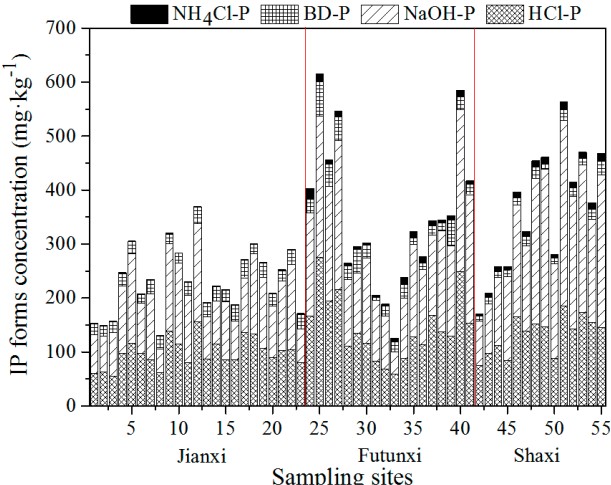

**Figure 6.** Concentration and spatial distributions of inorganic phosphorus (IP) forms in sediments. Note: The red line is the limit of sampling points in different watershed.

Among all samples, Samples No. 25 and No. 40, which were located in the suburb of Guangze County and Shunchang County, respectively, had the highest IP concentrations. These areas consisted of forestland, farmland, bamboo forest and residential areas. Samples No. 33 and No. 42, which were located near woodland and grassland and weaker human activities in Jianning County and Ninghua Country, respectively, had the lowest IP concentrations. These results further illustrate that land use type had a significant effect upon the accumulation of IP in sediments in the River Basin. Indeed, the IP accumulation in these sediments is positively related to human activities [13,19].

The IP/TP ratios (Figure 7) of the sediments in Jianxi, Futunxi and Shaxi were 55.06%, 66.53% and 67.75%, respectively. Meanwhile, the concentrations of IP (Figure 8) in these same sediments in Jianxi, Futunxi and Shaxi were 233.42 mg·kg$^{-1}$, 349.00 mg·kg$^{-1}$ and 364.54 mg·kg$^{-1}$, respectively.

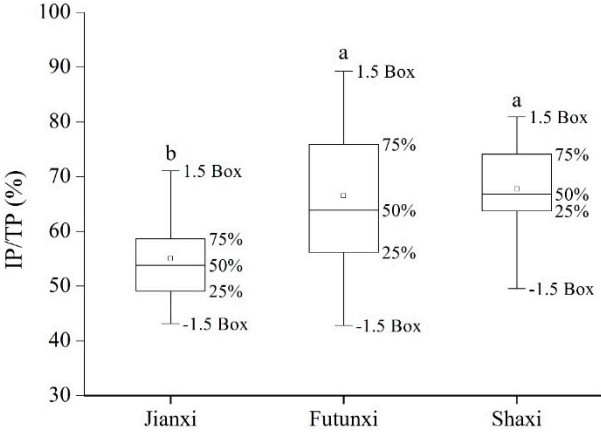

**Figure 7.** Distributions of IP/TP ratio in sediments. Different letters (a to b) implicate the significant difference between locations.

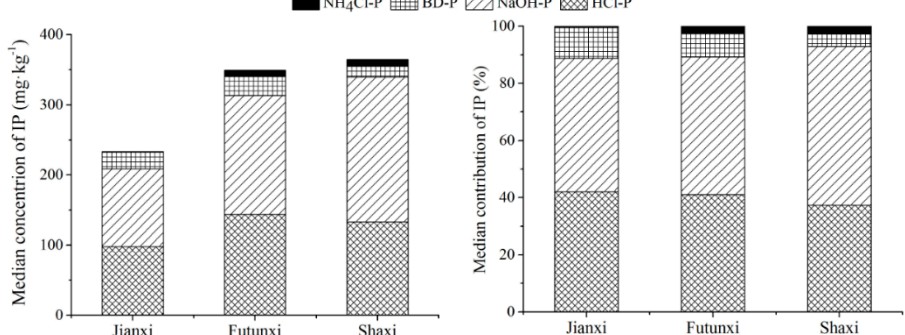

**Figure 8.** Contents and percentages of IP in sediments.

As shown in Figure 8, the concentrations of $NH_4Cl$-P (9.73 mg·kg$^{-1}$) and $NH_4Cl$-P /IP (2.68%) in sediments in the Shaxi Basin were higher than those in Futunxi (8.58 mg·kg$^{-1}$, 2.63%) and Jianxi (1.14 mg·kg$^{-1}$, 0.52%). The concentrations of TP, IP and $NH_4Cl$-P, which was a highly active component of IP with loose adsorbed phosphorus [19], were the maximum in sediments in the Shaxi Basin. Therefore, among all sediments in this study, the release potential of P in sediments in Shaxi Basin was the highest.

BD-P is redox-sensitive phosphorus consisting of phosphorus bound to iron hydroxides and manganese compounds. BD-P tends to have good mobility and bioavailability [13]. The average concentrations of BD-P in sediments in the Jianxi, Futunxi and Shaxi Basins were 23.52 mg·kg$^{-1}$, 27.20 mg·kg$^{-1}$ and 15.25 mg·kg$^{-1}$, respectively. The BD-P/IP ratios of sediments in Jianxi, Futunxi and Shaxi Basins were 10.75%, 8.09% and 4.38%, respectively.

NaOH-P and HCl-P are the primary components of IP in sediments. NaOH-P refers to phosphorus bound with aluminum and iron oxides, and is the main component of bioavailable phosphorus [13,19]. HCl-P is the most stable form of inorganic phosphorus, and is barely utilized by organisms. HCl-P consists of natural phosphorus, clastic apatite phosphorus, phosphorus combined with carbonate, and phosphorus produced by the hydrolysis of organic phosphorus. The NaOH-P/IP ratios in the sediments ranged between 32.88–63.30%. The average NaOH-P/IP ratios of Jianxi, Futunxi and Shaxi were 46.68%, 48.21% and 55.51%, respectively, which were significantly higher than the HCl-P/IP ratios (ranged between 31.09–51.55% with the averages of 42.05%, 41.06% and 37.43%, respectively). Previous studies have shown that NaOH-P and HCl-P were the main IP components in the sediments of rivers and lakes in China, although their contents may vary with locations. In southern China, granite and sandstone are the main parent rocks, resulting in high contents of NaOH-P; in northern China, limestone is abundant, and the content of HCl-P is usually higher than that of NaOH-P [26,27].

In summary, the average content of bioavailable phosphorus (BAP = $NH_4Cl$-P + BD-P + NaOH-P) in the sediments of the upper reaches of the Minjiang River was 195.88 mg·kg$^{-1}$, accounting for 64.85% of the total inorganic phosphorus. The average concentrations of BAP and the BAP/IP ratios in sediments in the Shaxi Basin were 231.69 mg·kg$^{-1}$ and 62.57%, respectively, which were significantly higher than that in Jianxi (135.61 mg·kg$^{-1}$, 57.95%) and Futunxi (205.23 mg·kg$^{-1}$, 58.94%). In summary, in the Shaxi Basin, the average concentrations of TP and BAP were the highest, and the ratio of BAP/IP in sediments was the largest. Therefore, the sediments in the Shaxi Basin exhibited highest risks of phosphorus release.

### 3.2. Factors Affecting Nitrogen and Phosphorus Forms in Sediments and the Environmental Significance

3.2.1. Factors Affecting Nitrogen Form and the Environmental Significance

According to the correlation between the basic physical/chemical indices of sediments and the nitrogen forms (Table 4), the nitrogen forms were not directly related to the pH value (which ranged from 4.5 to 6.3, and were thus acidic). Thus, the acidity and alkalinity fluctuation of the sediments within a certain range has weak impacts upon the nitrogen forms in the Basin in complicated environments.

**Table 4.** Relationships between basic physicochemical parameters and nitrogen forms in sediments.

| Parameter | IEF-N | WAEF-N | SAEF-N | SOEF-N |
|:---:|:---:|:---:|:---:|:---:|
| pH | −0.212 | −0.300 | 0.233 | 0.155 |
| TOC | 0.287 | 0.572 * | −0.476 * | 0.472 * |
| TN | 0.227 | 0.355 | −0.086 | 0.722 ** |
| TP | 0.287 | 0.411 | 0.476 | 0.272 |
| Al | −0.107 | 0.598 * | 0.073 | 0.347 |
| Si | −0.534 * | −0.393 | 0.597 | −0.450 * |
| Fe | 0.129 | 0.352 | 0.359 * | 0.282 |
| Mn | −0.216 | 0.106 | 0.432 * | 0.062 |
| Ca | 0.345 | 0.375 | −0.06 | −0.036 |
| Clay | −0.107 | 0.598 * | 0.073 | 0.347 |
| Silt | −0.504 * | −0.393 | 0.507 | 0.282 |
| Sand | 0.129 | 0.352 | 0.559 | −0.550 * |

Note: * $p < 0.05$; ** $p < 0.01$.

At $p < 0.05$, IEF-N was correlated with Si and silt by coefficients of −0.534 and −0.504, respectively. Previous studies have shown that Si primarily came from minerals and biomass, while IEF-N was most easily utilized by organisms [13,24].

In addition, the content of IEF-N was low and easily affected by various factors (salinity, acidity, hydrodynamics, temperature, organic matter, etc.). Meanwhile, the content of IEF-N fluctuated greatly, and was not directly correlated with other elements.

WAEF-N was positively correlated with TOC, Al and clay. To some extent, the increase of these three factors would facilitate the adsorption of WAEF-N on sediments. Specifically, the increase of TOC provided more effective adsorption sites for WAEF-N, while the increase of Al and clay led to an increase of the specific surface area, which was favorable to WAEF-N adsorption.

SAEF-N was positively correlated with Fe and Mn, and negatively correlated with TOC. Previous studies have shown that SAEF-N is primarily iron-manganese oxide bound nitrogen which is significantly affected by the redox conditions [12,18]. Specifically, SAEF-N tends to be released and accumulated under reduction and oxidation conditions, respectively. On the one hand, the increased contents of Fe and Mn in sediments led to an increased probability of combining with nitrogen, which in turn facilitated the formation of SAEF-N. On the other hand, the oxygen in the sedimentary environment was consumed, due to the mineralization of TOC, and the release of SAEF-N was positively related to the reducibility [18].

As the main bioavailable nitrogen in basin sediments, SOEF-N can reflect the level of organic nitrogen and the mineralization degree. Therefore, SOEF-N exhibited a significant positive correlation with TOC and TN in the studied watershed, with the correlation coefficients of 0.472 (at Level 0.05) and 0.722 ($p$ = 0.01), respectively. However, SOEF-N was negatively correlated with Si and sand at Level 0.05 (with the correlation coefficients of −0.450 and −0.550, respectively), while SAEF-N was positively correlated with Si and sand under certain conditions (with the correlation coefficients of 0.597 and 0.559, respectively). This may be attributed to the increasing contents of Si and sand. As the contents of Si and sand increased, the dynamic variation of water was exacerbated, and the fine grains such as clay were readily to be flushed away, resulting in an increased content of crude debris in sediments, void sizes and oxygen content. As a result, both the organic matter and SOEF-N were mineralized faster and then converted to SAEF-N components [13].

Four principal components (PC) accounting for 79.1% of the total variance were identified by factor loads (Table 5).

**Table 5.** Factor loads of nitrogen sources in sediments.

| Component | PC1 | PC2 | PC3 | PC4 |
|---|---|---|---|---|
| IEF-N | −0.104 | 0.319 | 0.326 | −0.568 |
| WAEF-N | 0.702 | 0.584 | 0.281 | −0.044 |
| SAEF-N | 0.251 | −0.355 | 0.721 | 0.741 |
| SOEF-N | 0.596 | 0.499 | 0.344 | −0.494 |
| TTN | 0.355 | 0.692 | 0.585 | −0.209 |
| pH | −0.087 | 0.035 | 0.076 | 0.213 |
| TOC | 0.603 | 0.213 | −0.052 | 0.012 |
| TN | 0.836 | 0.223 | 0.314 | 0.259 |
| TP | 0.306 | 0.581 | 0.662 | 0.215 |
| Al | 0.912 | 0.609 | 0.321 | −0.412 |
| Si | −0.370 | −0.505 | 0.119 | 0.634 |
| Fe | 0.424 | 0.082 | 0.726 | −0.516 |
| Mn | 0.500 | 0.291 | 0.430 | −0.410 |
| Ca | −0.345 | −0.475 | −0.036 | 0.306 |
| Clay | 0.449 | 0.714 | 0.329 | −0.022 |
| Silt | 0.295 | 0.523 | 0.253 | −0.326 |
| Sand | 0.273 | −0.538 | −0.121 | 0.513 |
| Characteristic value | 6.887 | 4.562 | 3.193 | 2.368 |
| Contribution rate of cumulative variance (%) | 30.8 | 52.5 | 67.7 | 79.1 |

The variance contribution rate of PC1 was 30.8%. The loads of WAEF-N, SOEF-N, TOC, TN and Al on PC1 were 0.702, 0.596, 0.603, 0.836 and 0.912, respectively. As a diagenetic element, Al was controlled by the geological background. Previous studies have shown that Al in sediments was mainly derived by minerals. Therefore, PC1 can be used as a natural source factor [24,28].

The variance contribution rate of PC2 was 21.7%. The loads of clay, silt and sand were 0.714, 0.523 and −0.538, respectively. The particle size classification factor can be used to assess PC2. Generally, the particles size classification in soil is influenced by its parent material and land use.

Thus, the particle size was the most representative factor of geological background and human activities [13,22]. In a specific area, the particle sizes in sediment can comprehensively reflect the land use type, which is characterized by different human activities. Therefore, compared with PC1, which is the main representative mineral source factor, PC2 can be regarded as the human interference factor related to land use and coverage change.

The variance contribution rate of PC3 was 15.2%, and the loads of SAEF-N, TTN, TP and Fe on PC3 were high. Specifically, the content of Fe in sediments is often used as an indicator of pollution, because it is greatly affected by the discharge of industrial and domestic wastewater along the river basin [13,19]. Therefore, PC3 can be considered as municipal wastewater, which has a great effect on the contents of SAEF-N, TTN and TP in sediments. It should be noted that from the TOC load value of

PC3, the change of TOC concentration in the sediments of the investigated area was almost unaffected by municipal wastewater. This phenomenon was consistent with the results of previous studies that the TOC in sediments of the studied area mainly came from the decaying of substances such as dead branches and deciduous leaves of higher plants by the erosion process of soil in the basin [15].

The variance contribution rate of PC4 was 11.4%, and the loads of IEF-N, SAEF-N, SOEF-N, Al, Si, Fe, sand and other factors were reasonable. According to the analysis above, the water body has strong dynamic variations, frequent biological activities, and consumes IEF-N components of organisms, resulting in an increased content of Si from biomass. Meanwhile, Al and Fe components adsorbed on clay and fine grains were flushed away, resulting in the increased contents of sand components, such as coarse debris, larger voids and the adequate content of oxygen in sediments. Thus, SOEF-N was mineralized into SAEF-N. Therefore, PC4 represented a water environmental factor that reflected biological quality and hydrodynamic effects [13,24,28].

In summary, the dominant influencing factors were the mineral sources in the basin, the land use types, the industrial and domestic wastewater discharge, the water biomass, and the dynamic environment.

### 3.2.2. Factors Affecting Phosphorus Form and the Environmental Significance

According to the correlation of the basic physicochemical indices of sediments and the phosphorus forms (Table 6), the pH value of sediments was positively and negatively correlated with the NaOH-P content (R = 0.451) and the HCl-P content (R = −0.370), respectively, indicating that the acidity of sediments can affect the ecological effectiveness of phosphorus.

**Table 6.** Relationships between basic physicochemical parameters and phosphorus forms in sediments.

| Parameter | $NH_4Cl$-P | BD-P | NaOH-P | HCl-P |
|---|---|---|---|---|
| pH | 0.255 | −0.248 | 0.451 | −0.370 |
| TOC | −0.332 | −0.317 | −0.197 | 0.452 ** |
| TN | 0.145 | 0.014 | 0.404 | 0.288 |
| TP | 0.371 | 0.574 * | 0.776 ** | 0.229 |
| Al | 0.556 * | 0.474 | 0.523 | −0.332 |
| Si | −0.545 * | −0.479 | −0.427 | 0.447 ** |
| Fe | 0.392 * | 0.483 * | 0.547 ** | −0.160 |
| Mn | 0.188 | 0.222 | 0.302 | 0.203 |
| Ca | 0.313 | 0.393 | 0.290 | 0.658 * |
| Clay | 0.556 * | 0.474 * | 0.523 ** | −0.532 * |
| Silt | 0.488 | 0.422 | 0.402 | −0.403 |
| Sand | −0.479 * | −0.545 * | −0.427 | 0.647 ** |

Note: * $p < 0.05$; ** $p < 0.01$.

This also explained the phenomenon that NaOH-P was the dominant form of phosphorus in most lakes and rivers in southern China, which was characterized by the acidic soil from the granite area.

Among all forms of phosphorus, $NH_4Cl$-P was positively correlated with Al, Fe and clay, with the correlation coefficients of 0.556, 0.392 and 0.556, respectively. In addition, $NH_4Cl$-P was negatively correlated with Si and sand, with the correlation coefficients of −0.545 and −0.479, respectively. The results showed that fine particles had a large specific surface area, and were easy to adsorb to $NH_4Cl$-P, Al and Fe components. The content of Si elements in coarse particles was higher, thus the adsorption capacity of the coarse particles to $NH_4Cl$-P, Al and Fe components was weaker. In addition, Si represented biomass to a certain extent. When the Si content was higher, more $NH_4Cl$-P was consumed (highest bioactivity). Therefore, $NH_4Cl$-P and Si were often reported to be negatively correlated [13].

BD-P is a redox-sensitive phosphorus that is readily released to water under reduction conditions. BD-P was positively correlated with TP, Fe and clay, and negatively correlated with sand. High specific

surface area of clay can harbor more secondary iron mineral phases. Thus, although clay was more prone to the reduction conditions which can eliminate Fe binding sites for P, it still had more sites than sand. Therefore, to some extent, particle size had more significant effect on the content of BD-P in sediments than redox conditions.

NaOH-P was positively correlated with TP, Fe and clay and negatively correlated with Si and sand. In this study, NaOH–P was the main component of TP, and its content was negatively related to the distance from the sampling site to the pollution source. Therefore, the content of NaOH-P can represent the content of TP [29]. Additionally, NaOH–P was closely related to the grain size of sediments, which was greatly affected by the adsorption performance.

HCl-P was primarily the calcium-bound phosphorus derived from authigenic components or clastic rocks. Thus HCl-P was positively correlated with TOC, Si, Ca and sand and negatively correlated with clay.

Table 7 shows the factor loads of the phosphorus sources in sediments. Three main components, i.e., PC1, PC2 and PC3, were identified. The variance contribution rates of PC1, PC2 and PC3 were 45.8%, 20.4% and 14.5%, respectively. The cumulative contribution rate of PC1, PC2 and PC3 was 80.7%.

**Table 7.** Factor loads of phosphorus sources in sediments.

| Component | PC1 | PC2 | PC3 |
|---|---|---|---|
| $NH_4Cl$–P | 0.836 | 0.797 | −0.550 |
| BD-P | 0.734 | 0.681 | −0.445 |
| NaOH–P | 0.798 | 0.611 | −0.379 |
| HCl–P | −0.336 | −0.219 | 0.652 |
| BAP | 0.850 | 0.705 | −0.424 |
| pH | 0.344 | 0.287 | 0.216 |
| TOC | 0.292 | −0.407 | 0.297 |
| TN | 0.309 | 0.384 | −0.283 |
| TP | 0.799 | 0.532 | 0.304 |
| Al | 0.506 | 0.618 | −0.415 |
| Si | −0.419 | −0.326 | 0.562 |
| Fe | 0.677 | 0.438 | −0.358 |
| Mn | 0.383 | 0.379 | −0.049 |
| Ca | −0.281 | −0.322 | 0.695 |
| Clay | 0.598 | 0.711 | −0.479 |
| Silt | 0.436 | 0.519 | −0.252 |
| Sand | −0.385 | −0.605 | 0.624 |
| Characteristic value | 6.891 | 5.128 | 3.061 |
| Contribution rate of cumulative variance (%) | 45.8 | 66.2 | 80.7 |

$NH_4Cl$-P, BD-P, NaOH-P, BAP, TP, Fe and clay have significant loads on PC1. Specifically, Fe is greatly affected by industrial sewage and municipal sewage. Typically, Fe is not an indicator of sewage pollution. However, due to the high forest coverage and a very low industrialization level in the studied watershed, the Fe-containing wastewater was mainly produced from municipal sewage.

In many studies, the content of Fe was considered as an effective indicator for the contents of bioavailable phosphorus and total phosphorus [13,19]. In addition, municipal sewage tended to carry fine particles, which were readily deposited in basin sediments [13]. Therefore, it can be concluded that urban sewage discharge had a significant impact on the total phosphorus content, bioavailable phosphorus content, and the three active forms (PC1, PC2 and PC3). In addition, PC1 was classified as the municipal wastewater.

The loads of clay, silt and sand in PC2 were 0.711, 0.519 and −0.605, respectively. Similarly, the influence of particle size was obtained. Compared with coarse particles, fine particles exhibited higher adsorption capability to bioavailable phosphorus. Specifically, Al, which was a diagenetic element, was controlled by geological background [24,28]. The classification of soil particle sizes was affected by parent materials of soil and land use types. Therefore, PC2 can be regarded as an

indicator for land use type, which reflected the common influences of geological background and human activities [22,24].

In addition, HCl-P, Si, Ca and sand indices were primarily involved in PC3. Their loads were 0.652, 0.562, 0.695 and 0.624, respectively. The results indicated that the composition played a dominant role in HCl-P, while the main influences of the composition on Si, Ca and sand were consistent. This phenomenon may be attributed to sand-washing activities or dynamic effects of water bodies. Specifically, when the sedimentary environment of water bodies was interfered with, more fine particles which had been adsorbed with active phosphorus components were flushed away, while the coarse sand grains and the inert calcium and phosphorus were left behind. Therefore, PC3 can be considered as a hydrodynamic impact indicator.

In summary, the contents of phosphorus in the sampled sediments were significantly affected by urban sewage, geological background and land use type. In some cases, the contents of phosphorus were also influenced by the water environmental factors, such as hydrodynamic disturbances.

### 3.3. Ecological Risk Assessment of Nutrient Elements in Sediments

### 3.3.1. Evaluations Based on Single Pollution Index

According to the evaluation results based on a single pollution index (Table 8), the TOC pollution index of sediments in the upper reaches of the Minjiang River ranged between 1.11–3.05, which was similar to that in the Minjiang River Estuary (ranging between 2.2 and 2.7). Therefore, all samples were classified as Grade II, indicating that their TOC pollution risks were low [7].

**Table 8.** Classification of the single pollution index in sediments (%).

| Basin | Items | Risk Level | | | |
|---|---|---|---|---|---|
| | | I | II | III | IV |
| Jianxi | TOC | 0 | 100 | 0 | 0 |
| | TN | 0 | 52 | 48 | 0 |
| | TP | 17 | 78 | 4 | 0 |
| Futunxi | TOC | 0 | 100 | 0 | 0 |
| | TN | 0 | 33 | 39 | 28 |
| | TP | 11 | 67 | 22 | 0 |
| Shaxi | TOC | 0 | 100 | 0 | 0 |
| | TN | 0 | 29 | 50 | 21 |
| | TP | 14 | 43 | 43 | 0 |

According to the risk assessment results, the TN risks in sediments in the Jianxi Basin were classified as Grade II or III, indicating a low and moderate pollution risk level, respectively. The sediment samples in Futunxi and Shaxi were classified as Grade II, III or IV. The samples of Grade II and IV accounted for 28% and 21% of the total samples, respectively, indicating the heavy pollution of TN in Futunxi. It has been reported that TN samples of estuarine sediments in the lower reaches of the Minjiang River were of Grade III or IV [7]. Therefore, it can be concluded that the Basin of the Minjiang River generally had moderate to severe pollution of TN.

The single pollution index of TP in sediment samples ranged from 0.32 to 1.20, and the risk grade ranged from Grade I to III. Specifically, 78% and 67%, respectively, of the samples in Jianxi and Futunxi were slightly polluted (grade II). On the other hand, most of the samples in the Shaxi Basin were slightly (43%) or moderately (43%) polluted. Therefore, TP pollution in the Shaxi Basin was more prominent than that in the other two sub-basins. Therefore, it can be concluded that the risk of TP pollution in the upper reaches of the Minjiang River was lower than that in the lower reaches of this same Minjiang River (the index ranged from 1.0 to 1.4, Grade III) [7].

### 3.3.2. Evaluations Based on the Biological Effectiveness Index

According to the evaluations results (Table 9) based on the bioavailability indices of N and P in those sediments of the upper reaches of the Minjiang River, the bioavailability index of nitrogen ranged between 0.63 and 3.99, and the risk level ranged from Grade I to IV. The bioavailability index of nitrogen in sediments in the Jianxi Basin was mainly Grade I (61%), while both the nitrogen bioavailability indices of sediments in the Futunxi (61%) and Shaxi (71%) Basins were mainly Grade II. Therefore, in the Futunxi Basin, the results on the bioavailability index of nitrogen (11% of total samples are of grade IV) indicated that there were high pollution risks. Thus in certain areas of the Futunxi Basin, there was a high risk of nitrogen release from sediments and a severe potential biological hazard related to nitrogen.

**Table 9.** Classification of bioavailability indices of nitrogen and phosphorus in sediments (%).

| Basin | Items | Risk Level | | | |
|---|---|---|---|---|---|
| | | I | II | III | IV |
| Jianxi | TN | 61 | 39 | 0 | 0 |
| | TP | 61 | 35 | 4 | 0 |
| Futunxi | TN | 22 | 61 | 6 | 11 |
| | TP | 50 | 33 | 17 | 0 |
| Shaxi | TN | 21 | 71 | 7 | 0 |
| | TP | 36 | 36 | 29 | 0 |

In this study, the bioavailability index of phosphorus in sediments ranged from 0.15 to 1.39, and the risk levels were mainly in Grade I and II. In other words, the sediment samples in the Jianxi and Futunxi Basins were risk-free (clean). Samples of Grade I accounted for 61% and 50% of the total samples in Jianxi and Futunxi, respectively.

In addition, the bioavailability index of phosphorus in sediments from the Shaxi Basin was homogeneously distributed in Grade I to III, i.e., the number of samples with negligible, mild and moderate pollution risks accounted for 36%, 36% and 29% of the total sediment samples, respectively. This was consistent with the evaluation results based on the single factor index of phosphorus in the Shaxi Basin described in the above paragraphs.

In summary, from the evaluation results based on the bioavailability index of nitrogen and phosphorus in sediments from the upper reaches of the Minjiang River, the pollution risks in the sediments of the Jianxi, Shaxi and Futunxi Basins were negligible, moderate and severe, respectively. The ecological risk of phosphorus in sediments was generally lower than that of nitrogen.

### 3.3.3. Comparison of the Two Evaluation Results

As shown in Figure 9, the pollution risks by the evaluation method based on biological effective indices of nitrogen and phosphorus were generally lower than those by the evaluation method based on a single pollution index. Specifically, when the evaluation method based on biological effective indices of nitrogen and phosphorus was used, the risk level of nitrogen was reduced from Grade III and IV to Grade I and II, and the risk level of phosphorus was reduced from Grade II, III and IV to Grade I. This result also indicated the high total amount of nitrogen and phosphorus in sediments of the target area. Therefore, the application of the biological effective index of nitrogen and phosphorus better reflects that the total amount of nitrogen and phosphorus in sediments in the study area is relatively high, but the bioavailability and release risk are relatively reduced, and the overall pollution level is not serious.

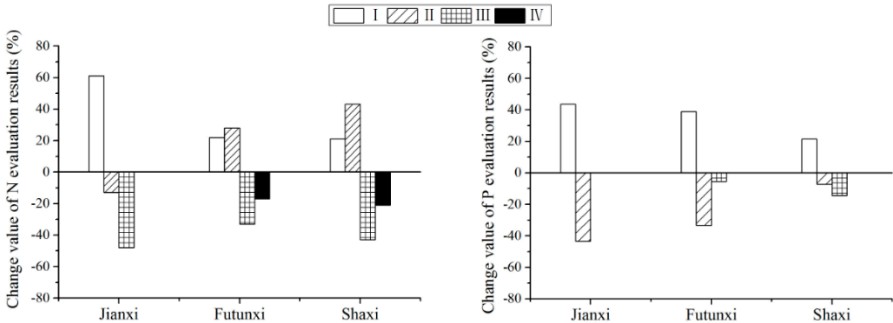

**Figure 9.** Change of evaluation results based on biological effective indices of nitrogen and phosphorus in sediments.

The investigated area is located in the mountain area of the upper reaches of the Minjiang River, thus the population and economy are underdeveloped, and the natural environment is good. In the studied area, forest land cover is high, rainfall is large, runoff is abundant, river bed slope is large, water flow is fast, water quality is good, and nutrient elements carried by the eroded sediment are not easy to accumulate in the river sediments. Therefore, considering both natural and social factors, the ecological threat of nitrogen and phosphorus elements in sediments in this area is small. The assessment method of nitrogen and phosphorus based on the bioavailability index considered the total amount and bioavailable ecological contents of nitrogen and phosphorus in sediments. Therefore, the proposed assessment method can reflect the comprehensive ecological risk of nitrogen and phosphorus in sediments more scientifically.

## 4. Conclusions

Based on the analysis of the total distribution and ratio of nutrient elements in sediments, the overall nutrient environment of sediments in the upper reaches of the Minjiang River were investigated, and the main influencing factors were analyzed. The nitrogen and phosphorus forms in sediments were thoroughly studied. The factors affecting the distributions of nitrogen and phosphorus in sediments, as well as their environmental significance, were investigated through main cause analysis. The evaluation method based on biological effective indices of nitrogen and phosphorus was proposed to evaluate the geochemical behavior and ecological risk of nitrogen and phosphorus in sediments. As a reference, the ecological risks of total carbon, nitrogen and phosphorus in the target area were assessed using the evaluation method based on a single pollution index. The main conclusions are as follows:

(1) According to the results on the total distribution and the ratios of nutrient elements in sediments in the upper reaches of the Minjiang River, Jianxi had the most abundant TOC, Futunxi had the most abundant TN, and Shaxi had the highest average content of TP. However, the main nutrient elements in sediments were originated from land, and thus were related to land use, such as the land use type and the intensity distribution of each type in the basin.

(2) In the sediments in the target area, the percentage of bioavailable nitrogen in total nitrogen ranged from 22.18 to 55.49%. In addition, the contents of different components were ordered from high to low as follows: SOEF-N > SAEF-N > WAEF-N > IEF-N. The percentage of inorganic phosphorus in total phosphorus ranged from 42.80 to 89.26% and the chemical component distributions were ordered as follows: NaOH-P > HCl-P > BD-P > NH$_4$Cl-P. The distribution pattern can be attributed to the natural mineral composition, land use types, urban sewage discharge, water biomass, and hydrodynamic disturbance in the basin.

(3) According to the evaluation results based on the single pollution index, the risk of TOC in sediments in the target area was relatively low, the risk of TN was light/moderate in most areas, but severe pollution in certain areas (Futunxi and Shaxi Basins), and the risk of TP was light. The obtained pollution risks using the evaluation method based on biological effective indices of nitrogen

and phosphorus were generally lower than those using the evaluation method based on single pollution index.

(4) In the future assessments of nutrient pollution in the sediments of river basins, it is suggested that not only the total amount and forms distribution of nitrogen and phosphorus, but also the changing characteristics of aquatic ecosystems should be considered. However, in current sediment toxicology research, one of the key and challenging points is how to screen the standard test organisms and determine effective biological test methods. Therefore, in the future, mature evaluation criteria of aquatic biological indicators should be used to quantitatively analyze the biological activity effects of different nitrogen and phosphorus forms, and to further verify and improve the proposed evaluation method based on the biological effectiveness index of sediment nitrogen and phosphorus.

Then the current risk assessment method system for nutrient elements in sediments can be effectively improved, and the comprehensive ecological effects can be explored more scientifically.

**Author Contributions:** H.Y. (Hongmeng Ye) and H.Y. (Hao Yang) are co-first authors; X.Y., N.H. and H.W. conceived and designed the experiments; T.H., C.H. and G.L. carried out the method and performed the analysis.

**Funding:** This research was supported by National Natural Science Foundation of China (grant nos. 41372354, 41673108, and 41773097), A Project of Fujian Provincial Department of Education (grant no. JAT170591), Fujian Science and Technology Innovation Platform Construction Project (grant no. 2017N2005), and Key Science and Technology Program of Nanping City (grant no. N2017T02).

**Acknowledgments:** The authors would like to express their appreciation for the anonymous reviewers and journal editor whose comments have helped to improve the overall quality of this paper.

**Conflicts of Interest:** The authors declare no conflict of interest.

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
