# Peer review of "Risk Assessment Based on Nitrogen and Phosphorus Forms in Watershed Sediments: A Case Study of the Upper Reaches of the Minjiang Watershed"

_sustainability, doi:10.3390/su11205565_

Round 1

Reviewer 1 Report

The topic of the paper is interesting.  However, the manuscript in this crucial parts is unclear or incomplete.In this form, the paper rather resembles a research report, not a scientific article.

The authors should carefully proofread and edit the manuscript for language and style.

Detailed comments:

Line 24-27 for a reader the connection between the Minjiang watershed and the Jianxi, Futunxi and  Shaxi basin is not clear at this part of the paper.

Line 24 MinJiang or Minjiang?

Introduction - the scientific background should be strongly underlined.

Materials and Methods section should be supplemented. The methodology and object of research must be clearly described. The article does not contain basic information about the studied area - climate, land use pattern, main sources of pollution.

Line 94 "Additionally, the upstream basin is a typical mountain river with three main tributaries, including Jianxi River, Futunxi River and Shaxi River." The basin is not a river.

Line 110 please change ~ to -.

Line 124 Master 2000 or Mastersizer 2000?

Line 182 All symbols in the formulas should be explained.

Discussion The article contains many general references to literature without specific values and names of the catchments where the quoted studies were conducted.

The article has not been prepared in accordance with the guide for MDPI authors.
Line 278 SAEF-N should appear in the materials and methods section.

Line 305 xxx???
In my opinion, the whole article lacks coherence and scientific quality.

Author Response

Responses to reviewers’ comments

Review 1 Report Form

Comments and Suggestions for Authors

The topic of the paper is interesting. However, the manuscript in this crucial parts is unclear or incomplete. In this form, the paper rather resembles a research report, not a scientific article.

The authors should carefully proofread and edit the manuscript for language and style.

Thank you for your valuable suggestion. We try our best to supplement the content and revise the presentation of the paper.

Detailed comments:

Line 24-27 for a reader the connection between the Minjiang watershed and the Jianxi, Futunxi and  Shaxi basin is not clear at this part of the paper.

R: Thank you for your suggestion. Now, we have added the contents “the Minjiang watershed (including Jianxi basin, Futunxi basin, and Shaxi basin).

Line 24 MinJiang or Minjiang?

R: Thank you for your comments. Here is “Minjiang”, and we have made corrections.

Introduction - the scientific background should be strongly underlined. 

R: Thank you for your advice. We made appropriate additions in the “Introduction”. For example, “Currently, grading criteria, which have been widely employed for sediment nutrient elements include the Sediment Quality Assessment Guidelines (by the Department of Environment and Energy Ontario, Canada), the single pollution index, the overall pollution index, the organic index and organic nitrogen index [5-11].”. “Specifically, scientific evaluation of the ecological risk of nutrient elements in sediments is particularly important for forestry mountainous basins due to their heterogeneity of land use and the complexity of topography and physiognomy.”.

Materials and Methods section should be supplemented. The methodology and object of research must be clearly described. The article does not contain basic information about the studied area - climate, land use pattern, main sources of pollution.

R: Thank you for your suggestion. we have added the relevant contents and the Figure 1 of land use in the “Materials and Methods” section.

Line 94"Additionally, the upstream basin is a typical mountain river with three main tributaries, including Jianxi River, Futunxi River and Shaxi River." The basin is not a river.

R: Thank you for your advice. Now, We have changed “the upstream basin” to “the upstream river” in the corresponding sentences.

Line 110 please change ~ to -.

R: Thank you for your comments. We have changed “~” to “-”.

Line 124 Master 2000 or Mastersizer 2000?

R: Thank you for your suggestion. We have make corrections “Master 2000” to “Mastersizer 2000 (Malvern, UK)”.

Line 182 All symbols in the formulas should be explained.

R: Thank you for your advice. Now, we have explained all symbols in the formulas.

Discussion The article contains many general references to literature without specific values and names of the catchments where the quoted studies were conducted.

R: Thank you for your comments. Because there are few reports in the research area, it is difficult to find the corresponding literature for comparison.

The article has not been prepared in accordance with the guide for MDPI authors.

R: Thank you for your suggestion. But, we can not find the guide for MDPI authors. Therefore we reference to published literature of “Sustainability”.

Line 278 SAEF-N should appear in the materials and methods section.

R: Thank you for your advice. Because different literatures have different abbreviations for nitrogen forms, they are unified here. “Nitrogen in iron manganese oxide form (SAEF-N)”  (instead of “nitrogen in iron manganese oxide form (IMOF-N)”) is appearing in the materials and methods section. And, We have changed “nitrogen in organic sulfide form (OSF-N)” to “nitrogen in organic sulfide form (SOEF-N)”.

Line 305 xxx???

R: Thank you for your comments. Now, we have added the expression of “weaker human activities” instead of “xxx” in the corresponding sentences.

Reviewer 2 Report

The English language usage needs to be revisited throughout the paper. Often it is not clear what the point of a sentence is, and some ideas are ambiguously expressed. This reduces the value of otherwise interesting work. 

In paragraph 2 of the introduction, the individual indices need reference citations. On line 63, it is not clear what you mean by “morphological distributions” – chemical species, maybe? Line 64, I don’t think you mean “criteria” – maybe “measures” or simply “indices” if you are referring to the list above. Line 68 – what is ‘ecological content”? – maybe you mean “ecologically bioavailable fraction”

The last paragraph is difficult to follow. In the first sentence, I think you are listing what you did – this needs to be made clearer. In the next sentence, if you are referring to existing indices, you need to provide citations. If you developed indices or risk ratings based on your analyses, you should say so. I think you use the work “index” when “concentration” might be more accurate – but it is not clear.

Section 1.1 is filled with facts and no citations, implying that all of the information contained in the section was generated by the authors – if this is the case, the methodology should be described and/or cited; if not, the section needs references.

Methods are not clearly described throughout

Lines108-9 by “mud collector” do you mean core sampler? What kind and manufacturer? Line 112 – what instrumentation/sources were used to record location, land use and activities? Line 113 – what does “dried naturally” mean?, how were impurities removed (by hand or sieving?), grinding how? What “cool ventilated environment”?

Section 1.2.2– it is not always clear what methods were used (not enough citations – or not in the right places) or what sample preparation was done. The fractionation of N and P is illustrated in Figure 2, which is clearer than the text, but the method of analysis of the individual fractions is not shown on the figure, and some of the abbreviations in the text don’t match the ones in the figure. This section might be clearer if it were rearranged to refer to the figure sooner, but more citations and information about sample handling are needed.

Reference 6 lacks publication details.

Paragraph starting at line 182 – I think you have mis-defined your variables: Cs is the standard concentration, Ci is the total N or total P and the others are measured fractions of the total, aren’t they?

Paragraph 189 it is not clear when you really mean index/indices and when you mean concentrations or factors

Line 200, I think you mean “comparing”, not “compared with” – and you should state if you are comparing the mean or max values. I think it would be useful to see some statistical analysis here so you can state if the differences between the watersheds are significant

Statement in lines 239/40 needs a citation and/or explanation.

Section 2.1.2 I don’t think a description of the conditions around the high and low TTN values is enough to ascribe cause and effect, although the influence of land use on water (and sediment) quality is well known. The dataset should allow a more formal spatial analysis of the data. It seems like there should be enough information here to do a statistical analysis of the effects of position in the watershed and land use, if this information is available for the whole area, and nitrogen levels.

Figure 4 needs a more complete legend, and the statistical significance of differences should be reported

Line 270 need to define what you mean by “highest activity” and “contribution rate” – these are concentrations. Do you mean most bioavailable, and having the highest exchange rate between the sediment surface and the water? Or contributes most to the biological pool? I can’t tell what you mean here.

Line 287 – do you mean “nitrogen release from water body” or from the sediment to the water body? Or are you talking about denitrification here? This is confusing.

Line 305 – what is meant by xxx?

Lines 308/9 – should expand on the last sentence – otherwise it doesn’t really connect with the previous

Paragraph beginning at line 349 – what is the range of pH? If it is not a broad range, I don’t think lack of correlation is very meaningful (same with other constituents in table 4)

Line 353 by “powder” I think you mean silt?

Line 355 doesn’t make sense –maybe you mean high Si is consistent with a biomass-enriched environment with a high IEF-N demand. I’m not sure what you are trying to say in the next sentence – There are no significant correlations with the factors mentioned - ?

Point on lines 364/5 needs a citation – which previous studies?

Line 372 – often shows a positive correlation? Give a citation if you are talking about other work. If just your own results are being discussed, then just say they showed a positive correlation

Lines 375/6 as you discuss the relationship with Si and sand you say it is both positively and negatively associated. This doesn’t make sense as written. The table shows only negative correlation, so the point needs expansion and clarification (you say positive correlation under certain conditions – what conditions? Where do these data come from?)

The second paragraph on page 2 states that the particle size distribution is primarily influenced by human activity – I guess there may be buildup of particles in areas between high flow events, but the primary factor influencing the bottom morphology must be flow rate – obviously influenced by human factors, but also geomorphology of the region.

Line 417 – the “effectiveness” of phosphorus is probably not what you mean here – likelihood that it is adsorbed/bound maybe. Need to expand on why this makes NaOH-P the dominant form in southern China

I can’t figure out what the point is in the third paragraph in section 2.2.2 – it needs clarification – I think it is that the high specific surface area of clay can harbor more secondary iron mineral phases, so despite the fact that clays are more prone to reducing conditions which can eliminate these sites, there are more Fe binding sites for P in clay than in sand.

446/7 – states Fe is an indicator of sewage contamination without a citation. Fe is a common element, so not a good indicator of the presence of sewage. Certainly the reducing conditions downstream of a sewage outfall can induce reducing conditions and solubilize sediment Fe (meaning a negative indicator of sewage in sediments?) – anyway, this point needs expansion. Fe oxide coatings are strong P sorbents accounting for the correlation with TP and bioavailable P

Section 2.3.3 – based on the “bioavailable index” method, the severity of impact will decrease in any case where the bioavailable nutrient concentration is less than the non-reactive content – so it will make the “good” samples look better and the “bad” samples look worse by definition.  If the idea is that only the bioavailable fraction will have an ecological effect (at least in the short to medium term), then a better adjustment might be to change the standard value, based on an assumed average or typical bioavailable fraction and calculate the bioavailable index as the concentration of bioavailable nutrient over the adjusted standard. This would avoid discounting the impact of labile nutrients in an environment with a larger non-reactive background.

From my first reading of the abstract, I was expecting to see an index that included TN, TP and TOC (multiple different parameters) such as has been done for heavy metals, as opposed to simple adjustment of single analyte risk groupings based on bioavailability.

This is an interesting dataset. I think the presentation and insights gained would be stronger with a more thorough statistical analysis – at least for the comparison of distribution of pollutants among the sub-watersheds) and a spatial analysis of the results. Spatial analysis might bolster some of the comments made about land use and watershed characteristics that are stated but not really backed up by data or citations. It could also strengthen the discussion about point sources (wastewater discharge). Use of a multi-pollutant index derived from your data would also be interesting.

Author Response

Responses to reviewers’ comments

Review 2 Report Form

Comments and Suggestions for Authors

The English language usage needs to be revisited throughout the paper. Often it is not clear what the point of a sentence is, and some ideas are ambiguously expressed. This reduces the value of otherwise interesting work. 

Thank you for your valuable suggestion.We try our best to supplement the content and revise the presentation of the paper.

Detailed comments:

In paragraph 2 of the introduction, the individual indices need reference citations. On line 63, it is not clear what you mean by “morphological distributions” – chemical species, maybe? Line 64, I don’t think you mean “criteria” – maybe “measures” or simply “indices” if you are referring to the list above. Line 68 – what is ‘ecological content”? – maybe you mean “ecologically bioavailable fraction”

R: Thank you for your comments. Now, we use the better wording advided by you. Now, We have changed “morphological distributions” to “chemical species”, “criteria” to “indices”, and “bioavailable ecological content” to “ecologically bioavailable fraction” in the corresponding sentences.

The last paragraph is difficult to follow. In the first sentence, I think you are listing what you did – this needs to be made clearer. In the next sentence, if you are referring to existing indices, you need to provide citations. If you developed indices or risk ratings based on your analyses, you should say so. I think you use the work “index” when “concentration” might be more accurate – but it is not clear.

R: Thank you for your advice. In order to make the expression clearer, we have added references and modified some expressions in the corresponding sentences, as follows, “Then, an index of the ecologically bioavailable fractions of nitrogen and phosphorus based on our analyses was proposed and used to assess the ecological risks of sediments. Results of evaluations by the previous single pollution index [6-7] and the proposed index were compared and the evaluation index was confirmed accordingly.”

Section 1.1 is filled with facts and no citations, implying that all of the information contained in the section was generated by the authors – if this is the case, the methodology should be described and/or cited; if not, the section needs references.

R: Thank you for your suggestion. We have supplemented the references to the cited information, and at the same time, we have supplemented the information of land use (Figure 1b), geological background, geographical environment, human activities, water conditions, and so on.

Methods are not clearly described throughout

R: Thank you for your comments. Now, we have explained all symbols in the formulas of “Materials and methods” section. At the same time, the details of sample collection and processing were further supplemented, including the steps of extracting nitrogen and phosphorus forms.

Lines108-9 by “mud collector” do you mean core sampler? What kind and manufacturer? Line 112 – what instrumentation/sources were used to record location, land use and activities? Line 113 – what does “dried naturally” mean?, how were impurities removed (by hand or sieving?), grinding how? What “cool ventilated environment”?

R: Thank you for your advice. We rearranged the expression “Herein, longitudes and latitudes of the sampling points, the administrative scope, the land use and human activities in local areas were recorded based on both GPS data and local record. These samples were transported in air-sealed plastic bags, separately freeze-dried, homogenized, ground and sieved to 0.25 mm. The sieved samples were stored at 4℃ in plastic bags.”

Section 1.2.2– it is not always clear what methods were used (not enough citations – or not in the right places) or what sample preparation was done. The fractionation of N and P is illustrated in Figure 2, which is clearer than the text, but the method of analysis of the individual fractions is not shown on the figure, and some of the abbreviations in the text don’t match the ones in the figure. This section might be clearer if it were rearranged to refer to the figure sooner, but more citations and information about sample handling are needed.

R: Thank you for your suggestion. The method of analysis of the individual fractions is added (“The extraction processes of TTN fractions are shown in Figure 2 (a). The supernatant fluid was employed to determine contents of NH4+-N and NO3--N using the methods reported elsewhere (Wang et al., 2009) [12]. Most of the averages in triplicate measurements were reasonable, which thus can be regarded as the final values.” And “The phosphorus extraction processes are shown in Figure 2 (b). Each phosphorus fraction was quantitatively assessed by the molybdenum blue / ascorbic acid method [13,15].), and we unified some of the abbreviations in the text for matching the ones in the figure 2.

Reference 6 lacks publication details.

R: Thank you for your comments. Now, we have replaced the relevant literature (“Alvarezguerra M, Viguri J R, Casadomartínez M C, et al. Sediment quality assessment and dredged material management in Spain: Part I, application of sediment quality guidelines in the Bay of Santander.[J]. Integrated Environmental Assessment & Management, 2010, 3(4):529-538.”).

Paragraph starting at line 182 – I think you have mis-defined your variables: Cs is the standard concentration, Ci is the total N or total P and the others are measured fractions of the total, aren’t they?

R: Thank you for your advice. Yes, Cs is the standard concentration, Ci is the total N or total P and the others are measured fractions of the total. In the “1.3.1 Risk assessment methodology” section, Pi is the single evaluation index or standard index, Ci is the measured concentration of evaluation factor i (i is TOC, TN or TP, here), and Cs is the standard concentration of evaluation factor i. Ki is the biological efficiency coefficient of factor i (i is N or P, here). KN is the biological efficiency coefficient of N, and KP is the biological efficiency coefficient of P. Pi, Ci and Cs are consistent with the above. CTN, CTTN, CTP, CBAP are the standard concentrations of TN, TTN, TP and BAP, respectively.

Paragraph 189 it is not clear when you really mean index/indices and when you mean concentrations or factors.

R: Thank you for your suggestion. Now, “In this study, the effects of the basic physical and chemical parameters on the forms of N and P were reflected by the correlation coefficients between the basic physicochemical indices concentrations and the forms concentrations of N and P in sediments.”

Line 200, I think you mean “comparing”, not “compared with” – and you should state if you are comparing the mean or max values. I think it would be useful to see some statistical analysis here so you can state if the differences between the watersheds are significant

R: Thank you for your comments. We have added some statistical analysis for difference analysis in Table 2 (different letters (a to b) implicate the significant difference between locations). Now, we can find that “Differences in the same elements in the sediments of the three sub-basins are presented, indicating the differences in the accumulation of elements between the basins; significant differences among different elements are observed, indicating that the sources of different elements are different.”.

Statement in lines 239/40 needs a citation and/or explanation.

R: Thank you for your advice. Now, we have added the ciation in the sentence (“This is related to the severe farmland pollution in the upstream basin and the severe municipal pollution in the downstream basin [15].”).

Section 2.1.2 I don’t think a description of the conditions around the high and low TTN values is enough to ascribe cause and effect, although the influence of land use on water (and sediment) quality is well known. The dataset should allow a more formal spatial analysis of the data. It seems like there should be enough information here to do a statistical analysis of the effects of position in the watershed and land use, if this information is available for the whole area, and nitrogen levels.

R: Thank you for your suggestion. Your suggestion is very meaningful. Land use information has been supplemented (Figure 1(b)). However, the specific analysis and statistics involve watershed scale issues, and the workload is large. At the same time, the space of this paper is limited, so there is no further supplement.

Figure 4 needs a more complete legend, and the statistical significance of differences should be reported

R: Thank you for your comments. Now, we have added significant differences in Figure 4 and Figure 7.

Line 270 need to define what you mean by “highest activity” and “contribution rate” – these are concentrations. Do you mean most bioavailable, and having the highest exchange rate between the sediment surface and the water? Or contributes most to the biological pool? I can’t tell what you mean here.

R: Thank you for your advice. Now, We changed our expression: “Previous studies have shown that IEF-N in sediments may be an indicator of nitrogen mobilities and their availability for plant uptake [12,18]. Indeed, IEF-N accounts for over 50% of nitrogen released from sediment to water [25]. Therefore, nitrogen in sediments in the Shaxi River basin is readily released.”.

Line 287 – do you mean “nitrogen release from water body” or from the sediment to the water body? Or are you talking about denitrification here? This is confusing.

R: Thank you for your suggestion. Here is from the sediment to the water body. In the paper, it is “Indeed the contribution of SOEF-N to nitrogen released from the sediment to the water body is only inferior to that of IEF-N [12,18,25].” .

Line 305 – what is meant by xxx?

R: Thank you for your comments. Now, we have added the expression of “weaker human activities” instead of “xxx” in the corresponding sentences.

Lines 308/9 – should expand on the last sentence – otherwise it doesn’t really connect with the previous

R: Thank you for your advice. Originally, this sentence is mainly intended to express the differences in the distribution trend of nitrogen and phosphorus fractions in sediments of river basins. Now, in order to avoid misunderstanding, we have deleted this sentence.

Paragraph beginning at line 349 – what is the range of pH? If it is not a broad range, I don’t think lack of correlation is very meaningful (same with other constituents in table 4)

R: Thank you for your suggestion. Here, the pH value is range from 4.5 to 6.3, and other constituents also have a range of variations. Similar research methods can be found in the relevant literature (eg. [13] and [19]), for more details.

Line 353 by “powder” I think you mean silt?

R: Thank you for your comments. Now, we have changed “powder” to “silt”.

Line 355 doesn’t make sense –maybe you mean high Si is consistent with a biomass-enriched environment with a high IEF-N demand. I’m not sure what you are trying to say in the next sentence – There are no significant correlations with the factors mentioned - ?

R: Thank you for your advice. Now, we have deleted this sentence (“As a result, more Si consistent is enriched and more IEF-N is consumed in the biomass rich reaches”).

Point on lines 364/5 needs a citation – which previous studies?

R: Thank you for your suggestion. Here, we have added the relative previous studies (“Previous studies have shown that SAEF-N is primarily iron-manganese oxide bound nitrogen which is significantly affected by the redox conditions [12,18].”).

Line 372 – often shows a positive correlation? Give a citation if you are talking about other work. If just your own results are being discussed, then just say they showed a positive correlation

R: Thank you for your comments. We just own results (“Therefore, SOEF-N shows a significant positive correlation with TOC and TN in the studied watershed, with correlation coefficients of 0.472 (at Level 0.05) and 0.722 (p=0.01), respectively.”).

Lines 375/6 as you discuss the relationship with Si and sand you say it is both positively and negatively associated. This doesn’t make sense as written. The table shows only negative correlation, so the point needs expansion and clarification (you say positive correlation under certain conditions – what conditions? Where do these data come from?)

R: Thank you for your advice. We are very sorry for that we made a mistake in our previous statement. Now we have corrected it (“However, SOEF-N was negatively correlated with Si and sand at Level 0.05 (-0.450 and-0.550), while SAEF-N was positively correlated with Si and sand under certain conditions (0.597 and 0.559).”).

The second paragraph on page 2 states that the particle size distribution is primarily influenced by human activity – I guess there may be buildup of particles in areas between high flow events, but the primary factor influencing the bottom morphology must be flow rate – obviously influenced by human factors, but also geomorphology of the region.

R: Thank you for your suggestion. It is true that there are some effects of hydrodynamic forces such as velocity, but this factor is relatively weak. This factor is attributed to PC4 in this paper.

Line 417 – the “effectiveness” of phosphorus is probably not what you mean here – likelihood that it is adsorbed/bound maybe. Need to expand on why this makes NaOH-P the dominant form in southern China

R: Thank you for your comments. Now, an appropriate explanation was given, as following, “This is also the reason why NaOH-P is the dominant form of phosphorus in most lakes and rivers in southern China, which is characterized by acidic soil from the granite area.”.

I can’t figure out what the point is in the third paragraph in section 2.2.2 – it needs clarification – I think it is that the high specific surface area of clay can harbor more secondary iron mineral phases, so despite the fact that clays are more prone to reducing conditions which can eliminate these sites, there are more Fe binding sites for P in clay than in sand.

R: Thank you for your advice. We are glad to have taken your suggestion and replaced it with your statement (“Since high specific surface area of clay can harbor more secondary iron mineral phases, Fe binding sites for P in clay exceeded those in sand, despite the fact that clays are more prone to reducing conditions, which can eliminate these sites.”) in the “2.2.2 Factors affecting phosphorus form and the environmental significance” section.

446/7 – states Fe is an indicator of sewage contamination without a citation. Fe is a common element, so not a good indicator of the presence of sewage. Certainly the reducing conditions downstream of a sewage outfall can induce reducing conditions and solubilize sediment Fe (meaning a negative indicator of sewage in sediments?) – anyway, this point needs expansion. Fe oxide coatings are strong P sorbents accounting for the correlation with TP and bioavailable P

R: Thank you for your suggestion. Now, we have made the following additions, “Specifically, Fe is greatly affected by industrial sewage and municipal sewage. Although Fe is typically not an indicator of sewage pollution, Fe-containing wastewater in the study watershed mainly comes from municipal sewage due to its high forest coverage and low industrialization level.”.

Section 2.3.3 – based on the “bioavailable index” method, the severity of impact will decrease in any case where the bioavailable nutrient concentration is less than the non-reactive content – so it will make the “good” samples look better and the “bad” samples look worse by definition.  If the idea is that only the bioavailable fraction will have an ecological effect (at least in the short to medium term), then a better adjustment might be to change the standard value, based on an assumed average or typical bioavailable fraction and calculate the bioavailable index as the concentration of bioavailable nutrient over the adjusted standard. This would avoid discounting the impact of labile nutrients in an environment with a larger non-reactive background.

R: Thank you for your comments. Your consideration is comprehensive and scientific. However, in the long run, labile nutrients or relatively stable forms of nitrogen and phosphorus need to be transformed into bioavailable forms before they can be utilized by organisms. Therefore, it is feasible to revise the total amount with the biological effective coefficient at present.

From my first reading of the abstract, I was expecting to see an index that included TN, TP and TOC (multiple different parameters) such as has been done for heavy metals, as opposed to simple adjustment of single analyte risk groupings based on bioavailability.

R: Thank you for your advice. You're quite right. Because the current experimental system is not particularly perfect, without considering more factors, the original intention of scientific research is to try a modified evaluation method of nitrogen and phosphorus forms. For a short while, it is very difficult for us to revise it according to your suggestion, but it enlightens us greatly in our later research. It also makes us further aware of the significance of this research work, and we are full of greater exploration interest in the follow-up research.

This is an interesting dataset. I think the presentation and insights gained would be stronger with a more thorough statistical analysis – at least for the comparison of distribution of pollutants among the sub-watersheds) and a spatial analysis of the results. Spatial analysis might bolster some of the comments made about land use and watershed characteristics that are stated but not really backed up by data or citations. It could also strengthen the discussion about point sources (wastewater discharge). Use of a multi-pollutant index derived from your data would also be interesting.

 R: Thank you for your many valuable suggestions. In the future, according to your ideal suggestions, we will improve the experimental indicators, screen "characteristic factors" (multi-parameters), and create a "multi-pollution index" evaluation system. For a short while, it is very difficult for us to revise it according to your suggestion. In a word, thank you for giving us the courage and strength to persevere further. Thank you again for your detailed suggestions for reviewing the manuscript and for your guidance on the overall paper and research.

Reviewer 3 Report

A more detailed description should be provided in the section - Analysis method of factors affecting distributions of nitrogen and phosphorus forms,

In the section - Overview of the Minjiang River, please complete the description concerning: geological background, geographical environment, land use, human activities and water conditions,

Line 305: should be verified - near woodland and grassland and xxx

Table 5 should be provided instead of 1, 2, 3, 4 - PC1, PC2, PC3 i PC4,

Table 6 should be provided instead of 1, 2, 3 - PC1, PC2 i PC3

It should be noted that in Tables 8 and 9 the values are given in %,

Author Response

Responses to reviewers’ comments

Review 3 Report Form

Comments and Suggestions for Authors

A more detailed description should be provided in the section - Analysis method of factors affecting distributions of nitrogen and phosphorus forms,

R: Thank you for your advice. We try our best to supplement more content in the “1.3.2 Analysis method of factors affecting distributions of nitrogen and phosphorus forms” section.

In the section - Overview of the Minjiang River, please complete the description concerning: geological background, geographical environment, land use, human activities and water conditions,

R: Thank you for your suggestion. We try our best to supplement more content in the “1.1 Overview of the Minjiang River” section.

Line 305: should be verified - near woodland and grassland and xxx

R: Thank you for your comments. Now, we have added the expression of “weaker human activities” instead of “xxx” in the corresponding sentences.

Table 5 should be provided instead of 1, 2, 3, 4 - PC1, PC2, PC3 i PC4,

R: Thank you for your advice. We have changed the “1, 2, 3, 4” to “PC1, PC2, PC3 i PC4” in Table 5.

Table 6 should be provided instead of 1, 2, 3 - PC1, PC2 i PC3

R: Thank you for your suggestion. We have changed the “1, 2, 3, 4” to “PC1, PC2, PC3 i PC4” in Table 6.

It should be noted that in Tables 8 and 9 the values are given in %,

R: Thank you for your comments. We have noted “%” in Tables 8 and 9.

Round 2

Reviewer 1 Report

The authors thoroughly improved the work in accordance with the reviewer's comments. The revised version has scientific value and is consistent with the journal's scope.

The work contains minor editorial errors i.e.:

Line 276 [5,7] please remove superscript

Figure 1 - Industry - please exchange for industry

Author Response

Responses to reviewers’ comments

Review 1 Report Form

Comments and Suggestions for Authors

The authors thoroughly improved the work in accordance with the reviewer's comments. The revised version has scientific value and is consistent with the journal's scope.

Thank you for your valuable suggestion and affirmation of this manuscript.

Detailed comments:

The work contains minor editorial errors i.e.:

Line 276 [5,7] please remove superscript

R: Thank you for your suggestion. Now, we have removed the superscript [5,7].

Figure 1 - Industry - please exchange for industry

R: Thank you for your comments. Now, we have exchanged “Industry” for “industry” in Figure 1(b). In addition, we have made Figure 1(b) more beautiful and clear.

Reviewer 2 Report

The English language was not improved in this iteration of the paper, although the authors did address the specific instances pointed out in my review that were difficult to interpret. In some cases, the edits are no more clear than the original.

The added section from lines 88-92 is still unclear: what is meant by “the evaluation index was confirmed accordingly”? Both indices are calculated using the same data so the results will be similar – that can’t be used as confirmation. The difference between the outcomes needs to be compared to some biological index to demonstrate the ecological significance of the newer method – especially if a more complex and expensive testing procedure is proposed.

Line 101 – annual rainfall? Or is it really runoff (less some fraction of water that infiltrates the soil)? If the latter, the method of estimation should be described.

Line 132 – need a citation for the Chinese Academy of Science data

Paragraph beginning at line 192 should go to the beginning of the methods section, and the quality control procedure should be described.

Line 217 – I think you mean “safe” concentration, or should it be “standard” – meaning Cs?

Line 247 – it would be clearer to say what you mean by “physicochemical indices concentrations”

The added text beginning at line 262 is still unclear (“different “ or “difference” shows up six times in this sentence – there must be a better way to state what you mean)

Figure 3 (and 6) would be better if it were clear which samples were in which basin

Line 325 – not significantly different from Shaxi according to your figure. Are the concentration differences significantly significant? Apart from in the two box & whisker plots, the statistical significance of differences in concentrations is still not indicated.

Line 383 I would reverse order of Futunxi and Jianxi to be the same as above (and in decreasing order) – or put them all in the same order as in the graph

Paragraph beginning line 387 – are the differences between Shaxi and Futunxi significant? (Should provide p values when “significant” differences are mentioned).

Line 416 – I am not sure I agree that Shaxi has a higher risk of P release than Futunxi, given that it has less of the more readily available forms of P

Line 478 – Interesting that TOC did not vary with domestic waste

Line 506 – fine grains don’t “prefer” NH4Cl-P, Al and Fe – if anything it is the other way around – greater available surface area for sorption – and sand is largely made of Si

The biological effectiveness index as presented here can only be seen as a slight modification of the single pollution index, given that it doesn’t distinguish between the different fractions of bioavailable nutrient content. If the point is to account for bioavailability, then the differences among the bioavailable fractions should also be accounted for in the calculation of the index. As currently calculated the largest fraction of the bioavailable P, which in the authors’ words is “barely utilized”, has the largest effect on the index.

This work would be a lot stronger if it were possible to compare the index results to an ecological effect to see which measurement, if either, actually reflects ecological impact better.

OK line 620 states that single pollutant index did a good enough job

Line 653 – I don’t think you can say the results were more “accurate” without producing evidence of ecological conditions at each of the sites to back up the assertion that one is better than another.

Author Response

Responses to reviewers’ comments

Review 2 Report Form

Comments and Suggestions for Authors

The English language was not improved in this iteration of the paper, although the authors did address the specific instances pointed out in my review that were difficult to interpret. In some cases, the edits are no more clear than the original.

Thank you for your valuable suggestion. We have tried our best to improve our writing and embellish our expression.

Detailed comments:

The added section from lines 88-92 is still unclear: what is meant by “the evaluation index was confirmed accordingly”? Both indices are calculated using the same data so the results will be similar – that can’t be used as confirmation. The difference between the outcomes needs to be compared to some biological index to demonstrate the ecological significance of the newer method – especially if a more complex and expensive testing procedure is proposed.

R: Thank you for your comments. Now, we have revised the statement in this article.

In addition, it is very scientific that you suggest adding biological indicators for comparison. However, our experimental design did not synchronously test biological indicators. In addition, the choice of biological indicators to compare will be more convincing, but also worthy of our consideration.

Line 101 – annual rainfall? Or is it really runoff (less some fraction of water that infiltrates the soil)? If the latter, the method of estimation should be described.

R: Thank you for your advice. Here is “annual runoff”. This data is obtained by reading the literature, so the corresponding references ([16]) are added in the article.

Line 132 – need a citation for the Chinese Academy of Science data

R: Thank you for your suggestion. We have supplemented the citation for the Chinese Academy of Science data.

Paragraph beginning at line 192 should go to the beginning of the methods section, and the quality control procedure should be described.

R: Thank you for your comments. Now, we have made appropriate amendments to your proposal and supplemented the references.

Line 217 – I think you mean “safe” concentration, or should it be “standard” – meaning Cs?

R: Thank you for your advice. Here is “standard”. Now, we have revised the statement in this article.

Line 247 – it would be clearer to say what you mean by “physicochemical indices concentrations”

R: Thank you for your suggestion. Now, we have listed the specific “physicochemical indices (eg. pH, particle size, TOC, TN, TP, Fe, Mn, Al, Si, and Ca) ” in the article.

The added text beginning at line 262 is still unclear (“different “ or “difference” shows up six times in this sentence – there must be a better way to state what you mean)

R: Thank you for your comments. We have tried our best to improve our writing and embellish our expression in the sentence (“The difference in the contents of the same element in the sediments from the three basins indicated the variation of element accumulation origin among the basins. Meanwhile, the difference in the content among different elements can reflect the distinction of the main element sources in the basin.”).

Figure 3 (and 6) would be better if it were clear which samples were in which basin

R: Thank you for your advice. We have supplemented the dividing lines of sampling points in the watersheds of Figure 3 (and 6) to better distinguish the sources of different sampling points.

Line 325 – not significantly different from Shaxi according to your figure. Are the concentration differences significantly significant? Apart from in the two box & whisker plots, the statistical significance of differences in concentrations is still not indicated.

R: Thank you for your suggestion. Since the concentration of TTN and IP is not listed separately in this paper, it is not easy to show the difference of TTN and IP concentration.

Line 383 I would reverse order of Futunxi and Jianxi to be the same as above (and in decreasing order) – or put them all in the same order as in the graph

R: Thank you for your comments. We have changed the order of expression of different basins according to your suggestion.

Paragraph beginning line 387 – are the differences between Shaxi and Futunxi significant? (Should provide p values when “significant” differences are mentioned).

R: Thank you for your advice. Now, we have deleted the word (“significant”).

Line 416 – I am not sure I agree that Shaxi has a higher risk of P release than Futunxi, given that it has less of the more readily available forms of P

R: Thank you for your suggestion. We have supplemented and explained the corresponding paragraph in this paper (“In summary, in Shaxi basin, the average concentrations of TP and BAP were the highest, and the ratio of BAP/IP in sediments was the largest. Therefore, the sediments in the Shaxi basin exhibited highest risks of phosphorus release.”), which is more convenient for readers to understand.

Line 478 – Interesting that TOC did not vary with domestic waste

R: Thank you for your comments. The main sources of TOC in the studied watershed are supplemented(“It should be noted that from the TOC load value of PC3, the change of TOC concentration in sediments of the investigated area was almost unaffected by municipal wastewater. This phenomenon was consistent with the results of previous studies that TOC in sediments of the studied area mainly came from the decaying of substances such as dead branches and deciduous leaves of higher plants by the erosion process of soil in the basin [15]. ”).

Line 506 – fine grains don’t “prefer” NH4Cl-P, Al and Fe – if anything it is the other way around – greater available surface area for sorption – and sand is largely made of Si

R: Thank you for your advice. Now, We changed our expression: “The results showed that fine particles had large specific surface area and were easy to adsorb to NH4Cl-P, Al and Fe components. The content of Si elements in coarse particles was higher, thus the adsorption capacity of the coarse particles to NH4Cl-P, Al and Fe components was weaker. In addition, Si represented biomass to a certain extent. When the Si content was higher, more NH4Cl-P was consumed (highest bioactivity). Therefore, NH4Cl-P and Si were often reported to be negatively correlated [13].”.

The biological effectiveness index as presented here can only be seen as a slight modification of the single pollution index, given that it doesn’t distinguish between the different fractions of bioavailable nutrient content. If the point is to account for bioavailability, then the differences among the bioavailable fractions should also be accounted for in the calculation of the index. As currently calculated the largest fraction of the bioavailable P, which in the authors’ words is “barely utilized”, has the largest effect on the index.

R: Thank you for your suggestion. At present, there are many methods to classify the forms of nitrogen and phosphorus in sediments, which are difficult to unify. It will lead to the complexity of the evaluation method if different forms are evaluated by further distinguishing the differences of bioavailability, and it will be difficult to popularize the biological evaluation method in the short term. In addition, HCl-P which is the most stable form of inorganic phosphorus and barely utilized by organisms, is not belong to bioavailable phosphorus (BAP, BAP = NH4Cl-P + BD-P + NaOH-P).

This work would be a lot stronger if it were possible to compare the index results to an ecological effect to see which measurement, if either, actually reflects ecological impact better.

R: I appreciate your suggestion very much. It is very scientific that you suggest adding biological indicators for comparison. However, our experimental design did not synchronously test biological indicators. In addition, the choice of biological indicators to compare will be more convincing, but also worthy of our consideration.

OK line 620 states that single pollutant index did a good enough job

R: Thank you for your advice. We are very sorry for the misunderstanding you have made in the previous statement. Now we have reorganized the logical relationship of language expression for the sentence (“The investigated area is located in the mountain area of the upper reaches of Minjiang River, thus the population and economy are underdeveloped, and the natural environment is good. In the studied area, forest land cover is high, rainfall is large, runoff is abundant, river bed slope is large, water flow is fast, water quality is good, and nutrient elements carried by the eroded sediment are not easy to accumulate in the river sediments. Therefore, considering both natural and social factors, the ecological threat of nitrogen and phosphorus elements in sediments in this area is small. The assessment method of nitrogen and phosphorus based on bioavailability index considered the total amount and bioavailable ecological contents of nitrogen and phosphorus in sediments. Therefore, the proposed assessment method can reflect the comprehensive ecological risk of nitrogen and phosphorus in sediments more scientifically.”).

Line 653 – I don’t think you can say the results were more “accurate” without producing evidence of ecological conditions at each of the sites to back up the assertion that one is better than another.

R: Thank you for your suggestion. We highly admire your rigorous scientific attitude. We have added suggestions for further research in the future with your suggestions and changed the expression in the paper ((3) and (4) in the “4. Conclusions”).

At last, we would like to thank you in particular for all your valuable suggestions for revision and guidance for this scientific research work.
